# PREFERENCE OPTIMIZATION VIA KEY-STEP ERROR EXPLORATION FOR MULTI-STEP REASONING IN LLMS

## ABSTRACT

Large Language Models (LLMs) have shown promising performance in various reasoning tasks, but still confront challenges when dealing with complex multi-step reasoning. Existing methods suggest using fine-grained preference signals to guide mathematical reasoning step by step. However, how to efficiently build a fine-grained preference dataset with correct and incorrect steps remains an open problem. To address this challenge, we propose an efficient method for preference optimization via **K**ey-step **E**rror **Exp**loration (KEEP). Unlike previous methods that rely on extensive sampling of whole responses or predefined perturbations, KEEP implements a more controllable and lightweight step-level preference data construction. Specifically, KEEP designs a key step identification strategy to simplify data construction by focusing on critical steps of the reasoning path. Moreover, KEEP proactively explores the underlying errors on the key steps and speculatively remains high-valued errors for controllability. By focusing on key step error exploration, KEEP addresses a crucial gap in the efficient construction of fine-grained preference datasets. Extensive experiments on models from **7B to 70B** show KEEP delivers up to a **9.5%** performance gain across **6** mathematical reasoning benchmarks while reducing data generation costs by up to **10x**. We further demonstrate KEEP's broad generality, showing strong performance on diverse domains including logic, code generation, and long-form QA across **8 distinct domains**. Moreover, our analysis indicates KEEP's potential for training process supervision reward models (PRMs), which could effectively advance mathematical reasoning evaluation frameworks.

## 1 INTRODUCTION

Today, LLMs have shown remarkable performances in many reasoning tasks, such as: coding (Chen et al., 2023; Liu et al., 2024b), tool usage (Schick et al., 2023; Patil et al., 2023), logical reasoning (Kojima et al., 2022; Lu et al., 2024a), and knowledge question answering (Madaan et al., 2022; Singhal et al., 2023). However, mathematical reasoning remains a cornerstone challenge for LLMs as it requires a complex and long-chain reasoning, a small mistake can deviate the reasoning path and lead to a wrong answer.

Among many efforts to improve LLM's reasoning ability (Luo et al., 2023; Lu et al., 2024b; Tang et al., 2024; Mitra et al., 2024; Chen et al., 2024a; Liu & Yao, 2024; Xu et al., 2024b; Li et al., 2024; Shao et al., 2024c; Xin et al., 2024; Ying et al., 2024; Shao et al., 2024b), Directed Preference Optimization (DPO) (Rafailov et al., 2024) becomes a popular choice due to its simplicity. Despite its successful applications on open-text generation, summarization, and chat benchmarks, many studies (Lou et al., 2024; Lai et al., 2024) point out that DPO offers minimal benefits to LLM's mathematical reasoning as it is optimized on the whole response and overlooks the key difference between the chosen and rejected responses, which eventually misleads preference optimization and hinders its application on long-chain reasoning. As a remedy, previous works propose the use of fine-grained preference signals to guide mathematical reasoning step by step (Lai et al., 2024; Chen et al., 2024b; Lu et al., 2024d; Xu et al., 2024a; Liu et al., 2024a). However, how to efficiently curate a fine-grained preference dataset is an open problem. A common practice is to employ a commercial LLM (*e.g.*, GPT4) to label the reasoning steps (Lai et al., 2024; Lightman et al., 2023), but the accuracy of LLM being a process reward model is still low (Zheng et al., 2024) and can introduce much noise in preference data. Some works (Chen et al., 2024b; Lu et al., 2024d) propose to employ repeated rollout/high-temperature sampling to collect the possible errors, while this is effective, the

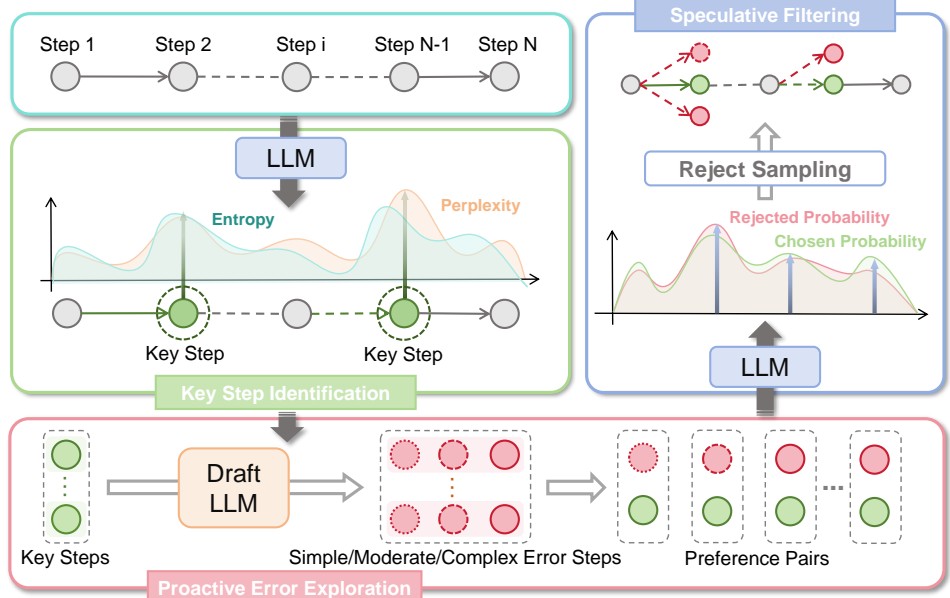

Figure 1: Overview of the KEEP framework. The process consists of three core stages: (1) **Key Step Identification**, which uses perplexity and entropy to locate critical reasoning steps; (2) **Proactive Error Exploration**, where a draft model generates plausible incorrect variants of these key steps; and (3) **Speculative Filtering**, which retains high-value, challenging preference pairs for optimization.

Table 1: Comparison of existing step-level preference construction methods. KEEP offers a compelling combination of efficiency, controllability, and realism.

| Method | Efficiency | Error Diversity | Controllability | Noise Risk | Limitation |
|---|---|---|---|---|---|
| High-temp / repeated rollout | ✗ | ✓ | ✗ | Low | High inference cost, poor scalability |
| LLM-as-PRM annotation | ✗ | ✓ | ✓ | High | Noisy labels due to imperfect PRM accuracy |
| Predefined error templates | ✓ | ✗ | ✓ | Low | Limited error types, unrealistic distribution |
| **KEEP (ours)** | ✓ | ✓ | ✓ | Low | — |

high inference cost hinders its scaling to large models. Yet a recent work (Xu et al., 2024a) imposes predefined errors to construct rejected responses, but its error type is limited by predefined prompts.

In this paper, we propose an efficient method for preference optimization via Key-step Error Exploration (KEEP), which implements a more controllable and lightweight step-level preference data construction. We summarize the distinctions between existing approaches and KEEP in Table 1. Moreover, Figure 1 illustrates the three main components of KEEP: 1) key step identification, 2) proactive error exploration, and 3) speculative filtering.

**Key Step Identification**. Along the long-chain reasoning path of math problem solving, LLMs can exhibit significant uncertainty on some key reasoning steps. To encourage the optimization to focus on these key steps, we employ the Perplexity (PPL) and information entropy to quantify the uncertain level of each step and identify the top-k steps as the key steps. PPL indicates the familiarity of LLMs with the current step, while information entropy reflects the step's informational content within the context. A higher PPL suggests a higher likelihood of errors, and higher entropy indicates greater contextual importance. We select steps with both high PPL and entropy as key steps.

**Proactive Error Exploration**. To address complex scenarios in real-world applications, we proactively explore potential errors for using a draft model, such as GPT-4o. For each key step, errors are classified as simple, moderate, or complex, ranging from character-level changes to adjustments throughout the reasoning process within the key step.

**Speculative Filtering**. To ensure that preference data is effective for LLMs, we propose Speculative Filtering based on rejection sampling to retain high-valued data. Specifically, for each keystep and its corresponding incorrect step, we examine their probabilities. We ultimately retain incorrect steps with high probabilities, as these are valuable for LLMs.

- **An efficient and novel framework**. We propose KEEP, a lightweight preference optimization method focusing on *key-step error exploration*, contrasting with prior works that use expensive rollouts or noisy annotations. KEEP achieves up to a **+9.5% accuracy gain** on the challenging AQUA benchmark and demonstrates broad generality, enhancing performance on tasks from **HumanEval** for code generation to long-form QA across **8 distinct domains**.

- **PPL–entropy-driven key-step identification**. We introduce a principled scoring rule that jointly leverages Perplexity (PPL) and information entropy. Its effectiveness is not just theoretically motivated but also empirically validated by a **strong positive correlation (Spearman's $\rho = 0.488$)** with the true expected improvement in Monte-Carlo simulations. This dual signal overcomes the bias of sampling methods that over-sample high-entropy steps while overlooking critical **high-PPL (high-complexity)** steps.

- **Multi-dimensional and robust experiments**. We conduct extensive evaluations on models from 7B to 70B across six mathematical datasets, including **competition-level benchmarks** like AIME24 and Odyssey-MATH. KEEP consistently outperforms strong baselines, including DPO-style method and RLVR method. Furthermore, we show KEEP can serve as a data engine to empower the RLVR ecosystem, training a PRM that boosts the crucial **error-step F1 score by 10+ points** on ProcessBench.

- **In-depth computational cost analysis**. Our theoretical and empirical analyses confirm that KEEP yields up to a **10x lower computational cost** compared to sampling-based approaches. Critically, its effectiveness is not reliant on expensive proprietary draft models; even when using a **7B open-source model**, KEEP **still significantly outperforms** strong baselines like SVPO and SCDPO.

- **Comprehensive and pedagogically-valuable error generation**. Through systematic analysis, we demonstrate that KEEP generates a more diverse set of errors, avoiding the **"mode collapse"** common in sampling-based methods. Moreover, our gradient-based attribution analysis confirms these generated errors are **pedagogically high-impact**, providing stronger and more meaningful learning signals to the model.

## 2 BACKGROUND

**Preliminary** Directed Preference Optimization (DPO) (Rafailov et al., 2024) becomes a popular choice due to its simplicity, which directly uses pair-wise preference data to optimize the policy model. Specifically, the input data includes an input prompt $x$, and a preference data pair $(y_{\mathrm{w}}, y_{\mathrm{l}})$, where $y_{\mathrm{w}}$ represents the better output response, and $y_{\mathrm{l}}$ represents the undesirable output response. DPO aims to maximize the probability of the better output $y_{\mathrm{w}}$ and minimize that of the undesirable output $y_{\mathrm{l}}$. The optimization objective is formulated as:

$$L_{\mathrm{DPO}}(\theta) = -\mathbb{E}_{(x, y_{\mathrm{w}}, y_{\mathrm{l}}) \sim D} \left[ \log \sigma \left( \beta \log \frac{\pi_\theta(y_{\mathrm{w}}|x)}{\pi_{\mathrm{ref}}(y_{\mathrm{w}}|x)} - \beta \log \frac{\pi_\theta(y_{\mathrm{l}}|x)}{\pi_{\mathrm{ref}}(y_{\mathrm{l}}|x)} \right) \right].$$

where $D$ denotes the pair-wise preference dataset, $\sigma$ is the sigmoid function, $\pi_\theta(\cdot|x)$ is the policy model being optimized, $\pi_{\mathrm{ref}}(\cdot|x)$ is the reference model that remains unchanged during training, and the hyperparameter $\beta$ controls the deviation from the reference model.

## 3 METHOD

As shown in Figure 1, the KEEP system comprises three main modules: 1) key step identification, 2) proactive error exploration, and 3) speculative filtering. For an LLM and a math question&answer, we first identify the most confusing step in the reasoning process of answer as the key step. Then, we proactively explore the underlying errors on the key steps via using a draft model to propose the varying degrees of errors. Finally, we filter out the less-possible errors via reject sampling and use the remaining high-valued ones for preference optimization.

### 3.1 KEY STEP IDENTIFICATION

To identify key steps that are confusing or uncertain for LLMs in multi-step reasoning, we use the perplexity (PPL) and information entropy to score each step. Perplexity evaluates the model's

predictive capability for the text within a step $S$, calculated as:

$$\text{PPL}(S) = \exp\left(-\frac{1}{N}\sum_{i=1}^{N}\log P_\theta(w_i \mid w_{1:i-1})\right)$$

Here, $N$ is the number of words $w_i$ in the step $S$, and $P_\theta(w_i \mid w_{1:i-1})$ is the model's predicted probability for the $i$-th word given the preceding words within the same step.

Information entropy measures the uncertainty of the current step within the broader context, calculated by considering the predicted probabilities of each word in the step:

$$H(S, C) = -\frac{1}{N}\sum_{i=1}^{N} P_\theta(w_i \mid c_i)\log P_\theta(w_i \mid c_i)$$

Here, $S$ represents the current step, $C$ represents the context, and $c_i$ denotes the context including all words before $w_i$, both within the current step and from prior steps. $P_\theta(w_i \mid c_i)$ is the probability of the $i$-th word given this context $c_i$).

To combine PPL and information entropy, we use a normalized linear combination:

$$\text{Score}(S) = \alpha \cdot \text{norm}(\text{PPL}(S)) + \beta \cdot \text{norm}(H(S, C))$$

where

$$\text{norm}(X_i) = \frac{X_i - \min(\{X_j\})}{\max(\{X_j\}) - \min(\{X_j\})}$$

Here, $X_i$ represents the value for a specific step $S$, and $\{X_j\}$ is the set of values across all steps within the current answer. Moreover, $\alpha$ and $\beta$ are weight coefficients, both defaulting to 1.

Finally, we select the top-scoring steps up to a certain percentage (default 50%) as key steps. This percentage can be adjusted according to your needs.

**Theoretical Motivation.** Our PPL-Entropy scoring rule is grounded in a principled theoretical analysis, detailed in Appendix D. The analysis reveals two key insights: correcting low-probability steps yields larger log-likelihood improvement (Theorem D.1), and steps with higher conditional entropy have a greater potential for information gain on final correctness (Theorem D.2). Our scoring rule, therefore, translates these theoretical principles into a practical and computationally efficient metric that approximates the true expected improvement. The effectiveness of this theoretically-grounded approach is empirically validated by our Monte-Carlo analysis, which confirms a statistically significant correlation ($\rho = 0.488$) between our score and the true expected improvement.

### 3.2 Proactive Error Exploration

For each identified key step, we proactively explore the underlying errors on the key steps to generate incorrect steps at three different levels: simple, medium, and complex.

- **Simple Errors**: These occur at the character level, such as spelling mistakes or symbol substitutions (e.g., replacing "+" with "-"). *Example*: For the mathematical step $2x + 3 = 7$, a simple error might be $2x + 3 = 8$.
- **Medium Errors**: These involve minor logical or calculation mistakes within a step. *Example*: In the step "subtract 3 from both sides to get $2x = 4$," a medium error might be "add 3 to both sides to get $2x = 10$."
- **Complex Errors**: These involve significant adjustments to the logic or reasoning process of the entire step. *Example*: Instead of solving the equation directly, a complex error might involve incorrectly applying a different method, such as using substitution when elimination is appropriate.

We input the key step and its context into a general-purpose draft model (e.g., GPT-4o) and use carefully crafted prompts to propose incorrect steps with these three-level possible errors. Detailed prompts can be found in Appendix A.2.

### 3.3 Speculative Filtering

To ensure that preference data is effective for LLMs, we propose Speculative Filtering based on rejection sampling to retain high-valued data. For each preference pair, we calculate the generation

probabilities for the correct step and the incorrect step using the LLM. These are referred to as $P_{\text{chosen}}$ and $P_{\text{rejected}}$, respectively. The formula is:

$$\text{P}(S, C) = \prod_{i=1}^{n} P_\theta(w_i \mid c_i)$$

where $S$ represents the current step, $C$ represents the context, and $c_i$ denotes the context including all words before $w_i$, both within the current step and from prior steps. $P_\theta(w_i \mid c_i)$ is the probability of the $i$-th word given this context $c_i$, and $n$ is the number of words in the step.

We use the ratio $\frac{P_{\text{rejected}}}{P_{\text{chosen}}}$ to determine whether to retain the current preference pair. A higher ratio indicates that the LLM finds it challenging to distinguish between the correct and incorrect steps, so we aim to retain such pairs to improve the model's discriminative ability.

### 3.4 Preference Optimization

Many studies (Lou et al., 2024; Lai et al., 2024) point out that traditional DPO offers minimal benefits to LLM's mathematical reasoning as it is optimized on the whole response and overlooks the key difference between the chosen and rejected responses, misleading the preference optimization and hindering its application on long-chain reasoning. Thus, we follow (Lai et al., 2024) to use fine-grained preference signals to guide mathematical reasoning step by step.

Specifically, given an input prompt $x$ and a sequence of initial correct reasoning steps $s_{1\sim k-1} = s_1, \ldots, s_{k-1}$, Step-wise DPO aims to maximize the probability of the next correct step $s_{\text{w}}$ and minimize the probability of the next incorrect step $s_{\text{l}}$. We let $z=(x, s_{1\sim k-1}, s_{\text{w}}, s_{\text{l}})$, and the optimization objective can be formulated as:

$$L(\theta) = -\mathbb{E}_{(z)\sim D}\left[\log \sigma\left(\beta \log \frac{\pi_\theta(s_{\text{w}}|x; s_{1\sim k-1})}{\pi_{\text{ref}}(s_{\text{w}}|x; s_{1\sim k-1})} - \beta \log \frac{\pi_\theta(s_{\text{l}}|x; s_{1\sim k-1})}{\pi_{\text{ref}}(s_{\text{l}}|x; s_{1\sim k-1})}\right)\right]$$

where $D$ is the dataset containing step-wise preference pairs, $\sigma$ is the sigmoid function, $\pi_\theta(\cdot|x; s_{1\sim k-1})$ is the policy model to be optimized, $\pi_{\text{ref}}(\cdot|x; s_{1\sim k-1})$ is the reference model that remains unchanged during training, $\beta$ is a hyperparameter that controls the distance from the reference model.

## 4 Experiments

To comprehensively validate our proposed method, we structure our experimental evaluation to systematically demonstrate KEEP's effectiveness, generality, and robustness. Our experiments are organized as follows:

- First, we establish KEEP's superior performance in **mathematical reasoning**, showing it consistently outperforms state-of-the-art DPO-style methods and is highly competitive with strong RLVR baselines (Section 4.2).

- Second, we demonstrate the **broad generality** of KEEP's core mechanism by evaluating it on diverse non-mathematical tasks, including logic, code generation, and long-form question answering (Section 4.3).

- Third, we conduct a series of **in-depth analyses and ablation studies** to validate each of our core components. These analyses confirm the robustness of our design choices and, critically, quantify the efficiency and quality of our data generation pipeline (Section 4.4).

- Finally, we explore KEEP's potential to **empower the RLVR ecosystem** by using its generated data to train high-quality Process Supervision Reward Models (PRMs) (Section 4.5).

To supplement these findings, we provide an extensive appendix that substantiates our core claims. This begins with the **theoretical motivation** for our key-step identification strategy (Appendix D), followed by deep empirical analyses, including: a detailed study confirming KEEP's ability to reduce data generation costs by up to **10x** (Appendix B); a rigorous quality audit of our generated data, which confirms its high purity (**<5% noise**) and pedagogical value via gradient attribution (Appendix F); and a comparative analysis showing KEEP produces a more diverse error distribution, avoiding the "mode collapse" seen in sampling methods (Appendix G). Further appendices delve into the roles of PPL and entropy (Appendix E) and provide illustrative case studies (Appendix H).

### 4.1 EXPERIMENTAL SETUP

#### 4.1.1 DATASETS

We evaluate our method on a wide array of reasoning benchmarks to test its effectiveness and generality. **1) For mathematical reasoning**, we use six datasets: three in-domain (GSM8k (Cobbe et al., 2021), MATH (Yu et al., 2023), and AQuA (Ling et al., 2017)) and three out-of-domain to test generalization (SVAMP (Patel et al., 2021), AIME24 (MAA, 2024), and Odyssey-MATH (Netmind.AI, 2024)). These benchmarks cover a wide spectrum of difficulty, from grade-school arithmetic to competition-level challenges. **2) For general reasoning**, we further test the broad generality of our method on benchmarks for logical reasoning (**ZebraLogic** (Mallen et al., 2023)), code generation (**MBPP** (Austin et al., 2021), **HumanEval** (Chen et al., 2021)), and multi-domain long-form question answering (**ELI5** (Fan et al., 2019)), where we evaluate across 8 distinct domains.

#### 4.1.2 BASELINES

- **General-purpose LLMs:** State-of-the-art open-source instruction-tuned LLMs like the Qwen2 series (Yang et al., 2024) and the Llama 3.1 series (Meta AI, 2024). We also report scores from leading closed-source models such as OpenAI's GPT-4o (OpenAI, 2024) and Anthropic's Claude-3.5-Sonnet (Anthropic, 2024) for context.
- **Mathematics-enhanced LLMs:** Models specifically optimized for mathematical reasoning, e.g., DeepSeekMath-RL (Shao et al., 2024b), Llemma (Azerbayev et al., 2024), ToRA (Gou et al., 2024), MAmmoTH (Yue et al., 2024a), and MathGenieLM (Lu et al., 2024c).
- **Stepwise DPO-optimized LLMs:** Advanced models fine-tuned with stepwise Direct Preference Optimization (DPO), such as Step-DPO (Lai et al., 2024), SVPO (Chen et al., 2024b), MCTS-DPO (Xie et al., 2024), SCDPO (Lu et al., 2024d), and RISE (Xu et al., 2024a).
- **RL with Verifiable Reward (RLVR) Methods:** We also include strong RL-based methods like PPO (Schulman et al., 2017) and GRPO (Shao et al., 2024a), which are mainstream approaches for mathematical reasoning.

### 4.2 SUPERIOR PERFORMANCE ON MATHEMATICAL REASONING

To validate KEEP's effectiveness, we first establish its performance in the challenging domain of math. This section is structured to demonstrate its capabilities, starting from common benchmarks and then scaling up to larger models and competition-level problems.

#### 4.2.1 MAIN
RESULTS ON COMMON BENCHMARKS

We begin by conducting comprehensive comparisons on four classic mathematical datasets. As shown in Table 2, our approach consistently achieves the best results among open-source models across both the Qwen2-7B and Llama-3.1-8B base models. Notably, our method not only delivers substantial improvements over the base models but also matches or surpasses the strongest baselines RISE, SCDPO and SVPO on key benchmarks like MATH and GSM8K, while establishing a clear advantage on AQuA. These gains are particularly significant. Specifically, KEEP boosts the Qwen2-7B model's score on the challenging MATH benchmark by **+7.7%** (from 52.2% to 59.9%), and improves the Llama-3.1-8B model's score on AQuA by a remarkable **+9.5%** (from 55.9% to 65.4%). This demonstrates the superior effectiveness of our fine-grained preference optimization strategy.

Table 2: Comparison of results on different commonly used mathematical datasets. † represents out-of-domain datasets.

| Model | GSM8K | MATH | AQuA | SVAMP† |
|---|---|---|---|---|
| Closed-source Models | | | | |
| GPT-4o | 96.0 | 78.1 | 82.2 | 94.3 |
| Claude-3.5-Sonnet | 94.9 | 68.5 | 77.5 | 92.9 |
| Open-source Models | | | | |
| Qwen2-7B-Instruct | 85.4 | 52.2 | 66.5 | 89.3 |
| Llama-3.1-8B-Instruct | 84.0 | 48.3 | 55.9 | 85.7 |
| DeepSeekMath-RL-7B | 87.7 | 52.7 | 59.0 | 88.4 |
| Llemma-7B | 36.4 | 18.0 | - | - |
| MathGenieLM-7B | 80.5 | 45.1 | - | 83.3 |
| MAmmoTH-7B | 53.6 | 31.5 | 44.5 | 67.7 |
| ToRA-7B | 68.8 | 40.1 | 23.6 | 68.2 |
| Step-DPO(7B) | 88.5 | 55.8 | 63.0 | 88.7 |
| SVPO(7B) | 81.7 | 59.5 | - | - |
| MCTS-DPO(7B) | 81.8 | 34.7 | - | - |
| SCDPO(7B) | 80.1 | 47.7 | 48.4 | 83.2 |
| RISE(Qwen2-7B) | 88.4 | **59.9** | 69.7 | 91.6 |
| RISE(Llama-3.1-8B) | 87.9 | 51.0 | 61.4 | 87.5 |
| KEEP(Qwen2-7B) | **89.1** | **59.9** | **72.8** | **91.9** |
| KEEP(Llama-3.1-8B) | 88.3 | 52.0 | 65.4 | 88.2 |

### 4.2.2 CHALLENGING RESULTS ON LARGE-SCALE MODELS

To test KEEP's scalability, we performed experiments on 70B+ scale models using more difficult, competition-level datasets. As shown in Table 3, our method continues to deliver consistent and meaningful improvements even on these challenging problems. For example, on the notoriously difficult **AIME24 benchmark, KEEP improves the score of Llama-3.1-70B from 7/30 to 9/30**, a substantial gain on problems designed to challenge human competitors. Similarly, on the Odyssey-MATH dataset, KEEP improves the performance of Llama-3.1-70B by +3.2%. This consistent performance on complex problems with SOTA models underscores the scalability and effectiveness of our approach.

Table 3: Results on competition-level datasets.

| Model | AQuA | AIME24[†] | Odyssey[†] |
|---|---|---|---|
| Closed-source Models | | | |
| GPT-4o | 82.2 | 3/30 | 52.9 |
| Claude-3.5-Sonnet | 77.5 | 4/30 | 48.0 |
| Open-source Models | | | |
| ToRA-70B | 41.3 | 0/30 | 26.8 |
| MAmmoTH-70B | 65.0 | 0/30 | 15.7 |
| Qwen2-72B-Instruct | 78.3 | 4/30 | 45.7 |
| Llama-3.1-70B-Inst. | 77.1 | 7/30 | 60.4 |
| Step-DPO(Qwen2-72B) | 77.5 | 4/30 | 50.1 |
| RISE(Qwen2-72B) | 79.1 | 4/30 | 49.4 |
| RISE(Llama-3.1-70B) | 77.7 | 7/30 | 58.9 |
| KEEP(Qwen2-72B) | **81.5** | 6/30 | 48.8 |
| KEEP(Llama-3.1-70B) | **80.7** | **9/30** | **63.6** |

### 4.2.3 FURTHER COMPARISON WITH SOTA METHODS

To provide a direct head-to-head comparison, we first applied core DPO-style baselines to the same 7B/8B base models. As presented in Table 4, the results confirm KEEP's strong and consistent performance, achieving superior or highly competitive results across the full suite of six mathematical datasets, validating its robustness and broad-ranging effectiveness within its paradigm. **Furthermore**, to situate KEEP against the mainstream RLVR paradigm, we conducted direct comparisons on key benchmarks. As shown in Table 5, KEEP achieves performance on par with or exceeding strong RLVR methods. Notably, on the challenging MATH benchmark, KEEP matches the strong PPO baseline while significantly outperforming GRPO. This suggests that our data generation strategy can produce training signals competitive with those from traditional RL pipelines in these reasoning tasks.

Table 4: Fair comparison with DPO-style methods using the same base models.

| Base model | Method | GSM8K | MATH | AQuA | SVAMP | AIME | Odyssey |
|---|---|---|---|---|---|---|---|
| | DPO | 86.1 | 49.1 | 70.9 | 89.8 | 2/30 | 34.0 |
| | SVPO | 86.7 | 52.2 | 71.3 | 89.6 | 0/30 | 12.6 |
| Qwen2-7B-instruct | SCDPO | 86.8 | 51.6 | 70.5 | 89.6 | 1/30 | 18.9 |
| | RISE | 88.4 | 59.9 | 69.7 | 91.6 | 2/30 | 36.2 |
| | **KEEP** | **89.1** | **59.9** | **72.8** | **91.9** | **4/30** | **44.4** |
| | DPO | 82.8 | 50.1 | 59.8 | 84.1 | 3/30 | 48.2 |
| | SVPO | 84.8 | 48.7 | 59.1 | 84.3 | 2/30 | 54.1 |
| Meta-Llama-3.1-8B-Instruct | SCDPO | 84.4 | 48.8 | 63.2 | 85.1 | 2/30 | 48.5 |
| | RISE | 87.9 | 51.0 | 61.4 | 87.5 | 3/30 | 49.3 |
| | **KEEP** | **88.3** | **52.0** | **65.4** | **88.2** | **4/30** | **54.3** |

### 4.3 BROAD GENERALITY ACROSS DIVERSE REASONING DOMAINS

A key advantage of KEEP is that its core mechanism—identifying and correcting errors at points of high model uncertainty—is domain-agnostic. To validate this, we applied KEEP to non-mathematical reasoning tasks. As shown in Table 6, KEEP brings significant improvements to logical reasoning and code generation, outperforming other DPO-style methods. For example, it improves the score on HumanEval from a 47.5 baseline to 51.6. **Furthermore**, we evaluated KEEP on a multi-domain long-form QA task (ELI5), where verifiable reward signals are often unavailable, making RLVR methods challenging to apply. As presented in Table 7, KEEP consistently improves answer quality across 8 diverse domains, enhancing not only overall scores but also specific dimensions like accuracy and completeness. These results strongly suggest that KEEP is a general framework for enhancing complex reasoning in LLMs.

### 4.4 IN-DEPTH ANALYSIS AND ABLATION STUDIES

#### 4.4.1 ABLATION ON CORE COMPONENTS

To demonstrate the importance of each module in our method, we designed an ablation study. As shown in Table 8, removing any of the three core components—key step identification, proactive

Table 5: Comparison with mainstream RLVR methods on key benchmarks, using Qwen2-7B-instruct as the base model.

| Method | GSM8K | MATH |
|---|---|---|
| Base Model | 85.4 | 52.2 |
| PPO (RLVR) | 87.5 | 59.4 |
| GRPO (RLVR) | **89.1** | 48.0 |
| **KEEP** | **89.1** | **59.9** |

Table 6: Performance on Logic and Code Tasks. MBPP and HumanEval are code benchmarks. Qwen2-7B-instruct as the base model.

| Method | ZebraLogic | MBPP | HumanEval |
|---|---|---|---|
| Base | 21.5 | 42.2 | 43.9 |
| DPO | 20.8 | 42.0 | 45.1 |
| RISE | 23.2 | 42.4 | 47.5 |
| **KEEP** | **35.8** | **43.1** | **51.6** |

Table 7: Multi-domain long-form QA (ELI5) results on 8 domains.

| Domain | Overall Score | | Accuracy Score | | Completeness Score | | Clarity Score | |
|---|---|---|---|---|---|---|---|---|
| | Baseline | KEEP | Baseline | KEEP | Baseline | KEEP | Baseline | KEEP |
| Biology | 7.48 | **7.75** | 7.53 | **8.12** | 6.66 | **6.89** | 8.29 | **8.50** |
| Chemistry | 7.45 | **7.63** | 7.82 | **8.11** | 6.49 | **6.86** | 8.08 | **8.10** |
| Economics | 7.67 | **7.81** | 8.14 | **8.31** | 6.77 | **7.15** | 8.05 | **8.23** |
| Engineering | 7.44 | **7.71** | 7.55 | **7.90** | 6.76 | **6.92** | 7.89 | **8.12** |
| Mathematics | 7.72 | **8.12** | 8.35 | **8.76** | 6.77 | **7.32** | 7.95 | **8.22** |
| Physics | 7.26 | **7.35** | 7.89 | **7.94** | 6.42 | **6.56** | 7.53 | **7.84** |
| Technology | 7.71 | **7.99** | 8.02 | **8.35** | 6.98 | **7.12** | 8.15 | **8.27** |
| Other | 7.32 | **7.46** | 7.60 | **7.71** | 6.55 | **6.69** | 7.88 | **7.94** |

error exploration, or speculative filtering—leads to a significant performance drop on both GSM8K and MATH datasets. This validates the integral role each component plays in KEEP's effectiveness.

Table 8: Ablation study of training settings.

| Method | GSM8K | MATH | Method | GSM8K | MATH |
|---|---|---|---|---|---|
| Qwen2-7B-Instruct | 85.4 | 52.2 | Llama-3.1-8B-Instruct | 84.0 | 48.3 |
| QWEN2-7B + KEEP | **89.1** | **59.9** | LLAMA-3.1-8B + KEEP | **88.3** | **52.0** |
| - w/o key step identification | 87.8 | 56.6 | - w/o key step identification | 83.8 | 43.8 |
| - w/o proactive error exploration | 88.0 | 58.3 | - w/o proactive error exploration | 84.8 | 50.9 |
| - w/o speculative filtering | 87.8 | 57.7 | - w/o speculative filtering | 86.2 | 50.7 |

### 4.4.2 PRACTICALITY WITH SMALL-SCALE DRAFT MODELS

A key concern for data generation pipelines is their reliance on powerful, often proprietary, draft models. We tested KEEP's practicality by using small, open-source QWen models (7B and 13B) as the draft model. Table 9 shows that even with a 7B draft model, KEEP consistently and significantly outperforms strong baselines like SVPO and SCDPO. This result is critical, as it demonstrates that KEEP's effectiveness stems from its structured data generation process, not the sheer power of the draft model, making it a practical and cost-effective solution.

Table 9: The results of experiments using smaller open-source models as Draft Model for KEEP.

| Method | Draft Model | GSM8K | MATH | AQuA | SVAMP | AIME | Odyssey |
|---|---|---|---|---|---|---|---|
| DPO | - | 86.1 | 49.1 | 70.9 | 89.8 | 2/30 | 34.0 |
| SVPO | - | 86.7 | 52.2 | 71.3 | 89.6 | 0/30 | 12.6 |
| SCDPO | - | 86.8 | 51.6 | 70.5 | 89.6 | 1/30 | 18.9 |
| **KEEP** | 7B | 88.3 | 56.6 | 72.1 | 90.9 | 3/30 | 36.2 |
| **KEEP** | 13B | 88.7 | 58.2 | 72.3 | 91.2 | 3/30 | 41.5 |
| **KEEP** | GPT-4o | **89.1** | **59.9** | **72.8** | **91.9** | **4/30** | **44.4** |

### 4.4.3 ROBUSTNESS OF DESIGN CHOICES

To address concerns about the heuristic nature of our design choices, we conducted experiments on two key parameters. First, as shown in the left part of Table 10, we replaced the linear fusion of PPL and entropy with alternative, theoretically-grounded functions (Multiplicative and Log-Sum-Exp). All methods yielded strong, comparable results, confirming our framework is robust and not tied to a specific fusion function. Second, the right part of Table 10 shows that our choice of a 50% key-step threshold provides a robust balance between performance and computational cost, with similar performance observed for 30% and 70% thresholds.

Table 10: Robustness analysis of key design: fusion method (left) and key step proportion (right).

| Method | GSM8K | MATH | | Proportion | GSM8K | MATH |
|--------|-------|------|---|-----------|-------|------|
| KEEP (Linear Fusion) | **89.1** | **59.9** | | 30% | 87.6 | 57.8 |
| KEEP (Multiplicative) | 88.9 | **59.9** | | 50% | 89.1 | **59.9** |
| KEEP (Log-Sum-Exp) | 88.4 | 59.1 | | 70% | **89.3** | 59.2 |

To demonstrate cross-model consistency, we performed a robustness analysis on both Qwen2-7B-Instruct and Llama-3.1-8B-Instruct. We conducted a grid search for the fusion weights $\alpha$ (PPL) and $\beta$ (Entropy) over the values $\{0.0, 0.3, 0.5, 0.7, 1.0\}$. As shown in Table 11, on *both* models, the balanced range $[0.3, 0.7]$ yields optimal performance with negligible variance. Setting $\alpha \approx \beta \approx 0.5$ is universally effective regardless of the model architecture.

Table 11: Effect of $\alpha$ and $\beta$ on KEEP Performance

| **Model** | $(\alpha, \beta)$ | **GSM8K** | **MATH** | **Avg** |
|-----------|-------------------|-----------|----------|---------|
| | (1.0, 0.0) PPL-only | 88.1 | 57.8 | 73.0 |
| | (0.0, 1.0) Ent-only | 87.9 | 57.5 | 72.7 |
| Qwen2-7B | (0.3, 0.7) | 88.8 | 59.4 | 74.1 |
| | **(0.5, 0.5) Default** | **89.1** | **59.9** | **74.5** |
| | (0.7, 0.3) | **89.1** | 59.6 | 74.4 |
| | (1.0, 0.0) PPL-only | 85.8 | 48.0 | 66.9 |
| | (0.0, 1.0) Ent-only | 85.5 | 47.5 | 66.5 |
| Llama-3.1-8B | (0.3, 0.7) | 88.0 | 51.3 | 69.7 |
| | **(0.5, 0.5) Default** | **88.3** | 52.0 | **70.2** |
| | (0.7, 0.3) | 87.9 | **52.1** | 70.0 |

## 4.5 EMPOWERING RLVR WITH HIGH-QUALITY PRMS

Beyond being a standalone preference optimization method, we investigated KEEP's potential to enhance the broader RLVR ecosystem. A high-quality Process-level Reward Model (PRM) is the cornerstone of many RLVR systems. We used the data generated by KEEP to train an open-source PRM (RLHFlow-PRM-Mistral-8B) and evaluated it on the ProcessBench benchmark. As shown in Table 12, the PRM trained on KEEP's data significantly outperforms baselines, especially in the crucial task of identifying error steps (e.g., boosting the correct step F1 score on the Omni-MATH benchmark by over 12 points, from 42.3 to 54.4). This demonstrates that KEEP serves as a powerful data engine capable of producing high-quality supervision signals that can directly benefit and empower mainstream RLVR pipelines.

Table 12: Training PRM based on KEEP data, evaluation results on ProcessBench dataset. The KEEP-trained PRM shows superior performance in identifying both error and correct steps.

| Model | GSM8K | | | MATH | | | OlympiadBench | | | Omni-MATH | | |
|-------|-------|---------|-----|-------|---------|-----|-------|---------|-----|-------|---------|-----|
| | error | correct | F1 | error | correct | F1 | error | correct | F1 | error | correct | F1 |
| PRM | 33.8 | **99.0** | 50.4 | 21.7 | 72.2 | 33.4 | 8.2 | 43.1 | 13.8 | 9.6 | 45.2 | 15.8 |
| PRM+SPO | 38.6 | 98.4 | 55.5 | 20.4 | 74.3 | 31.8 | 8.3 | 57.5 | 14.5 | 9.2 | 41.0 | 15.1 |
| PRM+SCDPO | 33.8 | 98.4 | 50.3 | **22.4** | 70.7 | 34.0 | 8.6 | 40.1 | 14.2 | 10.3 | 42.3 | 16.5 |
| PRM+KEEP | **43.5** | 96.4 | **59.9** | **22.4** | **79.8** | **35.0** | **8.8** | **63.1** | **15.4** | **10.4** | **54.4** | **17.5** |

## 4.6 QUANTITATIVE ANALYSIS: MITIGATING OOD RISKS VIA SPECULATIVE FILTERING

To quantitatively verify the effectiveness of speculative filtering and assess Out-of-Distribution (OOD) risks, we conducted a post-hoc analysis measuring the proportion of "Low Probability Errors" (proxies for high-OOD risk) in our dataset before and after filtering. We define high-OOD risk as errors where the policy model assigns a very low probability ($\tau$).

As shown in Table 13, the filtering process significantly reduces the presence of low-probability outliers. We observe that **over 91%** of the extreme OOD errors ($P < 0.1$) are eliminated. This confirms that KEEP effectively cleans the "Draft" distribution to match the "Policy" distribution.

Table 13: Reduction in High-OOD-Risk Errors After Filtering. Note: "Before" refers to the raw data from the Draft Model; "After" refers to the dataset used for training after Speculative Filtering.

| Threshold ($\tau$) | % Low-Prob (Before) | % Low-Prob (After) | Reduction |
|---|---|---|---|
| $P < 0.1$ | 2.3% | **0.2%** | 91.3% $\downarrow$ |
| $P < 0.2$ | 10.3% | 3.4% | 67.0% $\downarrow$ |
| $P < 0.3$ | 27.0% | 17.1% | 36.7% $\downarrow$ |

## 5 RELATED WORK

### 5.1 MATHEMATICAL REASONING

Large language models (LLMs) have demonstrated significant potential in mathematical reasoning. One major line of work leverages the chain-of-thought (CoT) reasoning framework and its extensions to elicit multi-step reasoning capabilities (Yao et al., 2024; Chen et al., 2024a; Tong et al., 2024; Yoran et al., 2023; Li et al., 2023). Another direction focuses on improving the supervised fine-tuning (SFT) process by curating higher quality or larger quantities of mathematical data (Yu et al., 2023; Yue et al., 2023; Liu & Yao, 2024; Lu et al., 2024b; Xu et al., 2024b; Li et al., 2024; Xin et al., 2024; Ying et al., 2024; Yue et al., 2024b; Tang et al., 2024; Mitra et al., 2024). Furthermore, to ensure the accuracy of calculations, many methods incorporate external tools like calculators and code interpreters (Cobbe et al., 2021; Shao et al., 2022; Gao et al., 2023).

### 5.2 PREFERENCE OPTIMIZATION

Preference optimization aims to improve the reasoning ability of LLMs by enabling them to generate higher-quality responses. For instance, methods based on Proximal Policy Optimization (PPO) (Lightman et al., 2024; Luo et al., 2023; Shao et al., 2024c) provide reward signals for entire generated outputs. However, these methods often involve complex training pipelines and depend heavily on the quality of a process supervision reward model (PRM). To simplify this, Direct Preference Optimization (DPO) (Rafailov et al., 2024) was proposed to learn directly from pairs of correct and incorrect responses.

Building on this, recent research has rapidly evolved towards fine-grained preference learning. One major direction is the **step-wise learning paradigm** (Lai et al., 2024; Lu et al., 2024d; Xie et al., 2024; Setlur et al., 2024; Xu et al., 2024a), which enables LLMs to learn from granular step-level feedback. A representative work in this vein is **Full-Step-DPO** (Xu et al., 2025), which introduces a promising holistic framework that leverages self-supervised process rewards to optimize the entire reasoning trajectory, effectively reducing reliance on external signals. Concurrently, a wave of pioneering approaches has emerged to explore finer granularity and uncertainty-aware mechanisms. For instance, **Selective-DPO** (Dong, 2025) proposes a sophisticated alignment strategy that prioritizes high-impact tokens based on reference model divergence. Similarly, **ConfPO** (Yoon et al., 2025) offers an efficient, model-free approach to identify preference-critical tokens via policy confidence. Expanding on the usage of uncertainty, **IUPO** (Li et al., 2025) introduces an insightful adaptive framework that strictly distinguishes between learning states to enhance performance in complex reasoning tasks. While these recent methods primarily focus on internal signals (e.g., confidence or probability) for sample weighting or token selection, **KEEP** distinguishes itself by introducing a complementary perspective: efficiently and actively exploring high-value potential errors via a draft model to construct diverse and high-quality step-wise preferences.

## 6 CONCLUSION

In this paper, we introduced KEEP, a novel framework to enhance LLM reasoning by efficiently generating high-quality, step-level preference data. Departing from expensive sampling, KEEP uses a lightweight process to identify critical reasoning steps, proactively explore targeted errors, and filter for the most valuable examples. Experiments demonstrate that this strategy significantly improves mathematical reasoning and generalizes robustly to diverse domains like code and logic. We conclude that KEEP offers a scalable and practical data generation paradigm, paving a more efficient path toward building reliable and capable reasoning models.

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

## A  Implementation Details

### A.1  Training Details

To ensure a fair comparison, we follow the training protocol used in Step-DPO, where mathematical problems are used for preference learning. The training dataset consists of approximately 10,000 problems, each paired with their corresponding step-by-step correct solutions. We use open-source models such as Qwen-2 (Yang et al., 2024) and Llama-3.1 (Meta AI, 2024) as the base LLMs and employ techniques like key step identification and error data filtering. Additionally, GPT-4o is used as the default draft model to generate more realistic incorrect steps for training.

The training process follows the standard Step-DPO methodology. 7B/8B models are trained for 4 epochs with a global batch size of 64. The parameter $\beta$ is set to 0.4. For 70B/72B models, we train for 4 epochs with LoRA and global batch size is set to 256, setting parameter $\beta$ to 0.5. We use DeepSpeed ZeRO3 with CPU offload to optimizes memory usage, enabling efficient training of massive models. The learning rate for all model training is set to 5e-7. AdamW optimizer and a cosine learning rate scheduler are used with warmup ratio set to 0.1.

### A.2  Details of Prompts

Table 14: The prompt for Proactive Error Exploration.

| **Prompt for Proactive Error Exploration**: |
| --- |
| Please read the following question, answer, and the key steps of the answer. Generate steps that appear possible correct but are actually incorrect for the keystep. The underlying incorrect steps should be derived from the context by altering the content to make it wrong. Note that you need to explore 3 incorrect steps for the keystep, each representing a different level of error: simple, medium, and complex. 
 * Simple represents minor errors: altering or disrupting a few characters compared to the correct step. 
 * Complex represents major errors: involving significant modifications and disruptions to the problem-solving thought process, logic, or reasoning. 
 * Medium falls between the two. 
 **Question**: 
 {question} 
 **Answer**: 
 {full answer} 
 **key_step**: 
 {index of key step} 
 Generate the final incorrect steps according to the error tags, and make sure not to explain how the step is incorrect within the step itself. Please output in JSON format, strictly following the sequence, without omitting any part of the content, including descriptions and formulas. 
 Format: 
 {"error_1_tag":"Simple","error_1_"+str(key_step_x)+"":"...","error_2_tag":"Medium","error_2_"+str(key_step_x)+"":"...", "error_3_tag":"Complex", "error_3_"+str(key_step_x)+"":"..."} 
 Not include extra characters in the output. |

## B  Analysis of Computational Costs

In this section we discuss the cost of KEEP and the *sota* method, i.e., StepDPO, and try to explain our conclusion from a perspective of the average consumption of tokens. Before the detailed proof, we introduce the two language models involved in the two methods: the teacher model and the target model. The teacher model is a model that has hundreds of billions of parameters such as GPT-4o and we utilize it for automatic error correction of data. The target model is the model we aim to train and its number of parameters can range from 0.5 billion to 70 billion or even bigger. Thus the average consumption of tokens of the two models should be considered separately. Then we introduce the notations in this section. Both KEEP and StepDPO involve a existing Q-A dataset $\mathcal{Q} = \{\mathbf{q}_i, \mathbf{a}_i^c\}_i^N$, where $N$ is the number of Q-A pairs in this dataset. Operation $|\cdot|$ represents the length of a sentence, which can be a question in $\mathcal{Q}$ or output of a model. In the followings, we show that our method reduces token consumption for both the target model and the teacher model.

### B.1 ANALYSIS OF STEPDPO

We first analyze the three stages of StepDPO: **Sampling of Incorrect Answers**, **Error Step Localization**, and **Sampling of Correct Answers**. Sampling of Incorrect Answers and Sampling of Correct Answers involve the target model while Error Step Localization asks the teacher model to localize the error step. In the process of Sampling of Incorrect Answers, for a given question $\mathbf{q}_i$ in $\mathcal{Q}$, we input $\mathbf{q}_i$ and ask the model to answer the question step by step. Then, answers with a correct final result are collected. Suppose that the model has the probability $p_w$ of giving an incorrect answer of $n_i$ steps: $\mathbf{a}_i = \{s_1^i, s_2^i, ..., s_{n_i}^i\}$, where $s_k^i$ denotes the $k$-th step of the answer. Thus, in this process, the number of input tokens consumed is

$$\sum_{i=1}^{N} |\mathbf{q}_i|, \tag{1}$$

while the number of output tokens consumed is

$$\sum_{i=1}^{N} |\mathbf{a}_i| = \sum_{i=1}^{N} \sum_{k=1}^{n_i} |s_k^i|. \tag{2}$$

And $p_w N$ answers are collected. We denote $\mathcal{I} = \{i : \text{the answer of } \mathbf{q}_i \text{ is collected}\}$. Then in the process of Error Step Localization, for each answer in the collected $p_w N$ incorrect answers, the teacher model is asked to analyze the correctness of the answer step by step until it finds the first specific incorrect step. Since the question of the answer is also provided, the number of input tokens consumed is

$$\sum_{i \in \mathcal{I}} |\mathbf{q}_i| + |\mathbf{a}_i| = \sum_{i \in \mathcal{I}} |\mathbf{q}_i| + \sum_{i \in \mathcal{I}} \sum_{k=1}^{n_i} |s_k^i|. \tag{3}$$

Suppose that the first $r_i - 1$ steps are correct and the $r_i$-th step is incorrect. Thus, the number of output tokens consumed is

$$\sum_{i \in \mathcal{I}} \sum_{k=1}^{r_i} m_k^i, \tag{4}$$

where $m_k^i$ is the length of the analysis of $k$-th step of answer $\mathbf{a}_i$. Thus, in this process, we get a set of questions, correct steps, and incorrect steps $\mathcal{D} = \{\mathbf{q}_i, s_{1 \sim r_i - 1}^i, s_{r_i}^i\}_i$. Finally, in the process of Sampling of Correct Answers, for each sample $\mathbf{d}_i = (\mathbf{q}_i, s_{1 \sim r_i - 1}^i, s_{r_i}^i)$ from $\mathcal{D}$, we prompt the target model with the question $\mathbf{q}_i$ and the correct steps $s_{1 \sim r_i - 1}^i$ to complete the answer. The completed answer can be expressed as $\hat{\mathbf{a}}_i = \{s_{1 \sim r_i - 1}^i, \hat{s}_{r_i}^i, \hat{s}_{r_i + 1 \sim n_i'}^i\} = \{\hat{s}_1, \hat{s}_2, ..., \hat{s}_{n_i'}\}$, where $n_i'$ is the number of steps of the completed answer. Then we select the answers by checking the final result of the completed answer $\hat{\mathbf{a}}_i$. If correct, we construct a preference pair $(\mathbf{q}_i, s_{1 \sim r_i - 1}, s_{r_i}, \hat{s}_{r_i})$. Suppose that the model has the probability $p_r$ of giving a correct answer. Consequently, approximately $p_w p_r N$ preference pairs are collected. Thus, in this process, the number of input tokens is

$$\sum_{i \in \mathcal{I}} |\mathbf{q}_i| + \sum_{k=1}^{r_i - 1} |s_k^i|, \tag{5}$$

where $t_i$ is number of trials to get a correct answer. Clearly, the number of output tokens is

$$\sum_{i \in \mathcal{I}} \sum_{k=r_i}^{n_i'} |\hat{s}_k^i|, \tag{6}$$

Therefore, we have discussed the tokens consumption of the three stages of StepDPO. Now we analyze the number of overall tokens consumption of the target model and the teacher model. With (1), (2), (5), and (6), noticing that $s_k^i = \hat{s}_k^i, i < r_i$, the overall tokens consumed of the target model is

$$T_1^s = \sum_{i=1}^{N} |\mathbf{q}_i| + \sum_{i=1}^{N} \sum_{k=1}^{n_i} |s_k^i| + \sum_{i \in \mathcal{I}} (|\mathbf{q}_i| + \sum_{k=1}^{n_i'} |\hat{s}_k^i|). \tag{7}$$

With (3) and (4), the overall tokens consumed of teacher model is

$$T_2^s = \sum_{i \in \mathcal{I}} |\mathbf{q}_i| + \sum_{i \in \mathcal{I}} \sum_{k=1}^{n_i} |s_k^i| + \sum_{i \in \mathcal{I}} \sum_{k=1}^{r_i} m_k^i. \tag{8}$$

## B.2 ANALYSIS OF KEEP

Next we analyze the three stages of KEEP: **Key Step Identification**, **Proactive Error Exploration**, and **Speculative Filtering**. In the process of Key Step Identification, we take full advantage of $\mathcal{Q}$. For given question $\mathbf{q}_i$ and answer $\mathbf{a}_i^c$, which usually consists of several steps, we first select key steps of high perplexity and high information entropy. Only a single forward pass is required throughout the computational process. Let $n_i$ be the number of steps of $\mathbf{a}_i^c$ and $\mathbf{a}_i^c = \{l_1^i, l_2^i, ..., l_{n_i}^i\}$. In this process, the number of tokens consumed is

$$\sum_{i=1}^{N} |\mathbf{q}_i| + |\mathbf{a}_i^c| = \sum_{i=1}^{N} |\mathbf{q}_i| + \sum_{i=1}^{N} \sum_{k=1}^{n_i} |l_k^i|, \tag{9}$$

and for each Q-A pair $(\mathbf{q}_i, \mathbf{a}_i^c)$, $n_i^c$ key steps are selected. Then in **Proactive Error Exploration**, for each collected key step across all samples, we prompt the teacher model, called **Draft Model** in KEEP actually, to output lots of incorrect answers corresponding to each key step. Specifically, taking the answers of key steps and the question as reference, the Draft LLM is asked to output $M$ incorrect answers for correct reference answer of each key step at a time. Therefore we have got lots of preference pairs. The number of answers we asked the Draft LLM to output, $M$, is a little bigger fixed number like 3. Let $\mathcal{I}_i^c = \{k : l_k^i \text{ is a key step in } \mathbf{a}_i^c\}$. Clearly, $\mathcal{I}_i^c$ has $n_i^c$ elements. Thus, in this process, the number of input tokens is

$$\sum_{i=1}^{N} |\mathbf{q}_i| + \sum_{i=1}^{N} \sum_{k \in I_i^c} |l_k^i|, \tag{10}$$

and the number of output tokens is

$$\sum_{i=1}^{N} \sum_{m=1}^{M} \sum_{k \in I_i^c} |l_{km}^i|, \tag{11}$$

where $l_{km}^i$ one of the $M$ incorrect step output by the Draft Model. And we get nearly $NM$ preference pairs in this process. Notice that not all preference pairs we collect are proper because they are some kind of out-of-distribution data. Thus we need to conduct rejection sampling to select the high quality preference pairs in Speculative Filtering. Specifically, for the positive and negative instances within a preference pair, we calculate the probability that the LLM to be trained outputs the instance (correct or incorrect), respectively. Only those preference pairs for which the model demonstrates marginal probability differentials between positive and negative instances are considered high-quality and subsequently selected. Suppose the rejection rate $1 - p_a$, which is decided by the Draft LLM 's and LLM to be trained 's distributions. With the rejection rate $1 - p_a$ considered, the actual consumption of tokens is:

$$\sum_{i=1}^{N} \sum_{m=1}^{M} \sum_{k \in I_i^c} (|\mathbf{q}_i| + |l_{km}^i|), \tag{12}$$

and we eventually collected $p_a M \sum_{i=1}^{N} n_i^c$ preference pairs.

Therefore, in KEEP, the consumed tokens of target model is, with (9) and (12),

$$T_1^k = \sum_{i=1}^{N} (1 + M n_i^c)|\mathbf{q}_i| + \sum_{i=1}^{N} \sum_{k=1}^{n_i} |l_k^i| + \sum_{i=1}^{N} \sum_{m=1}^{M} \sum_{k \in I_i^c} |l_{km}^i|, \tag{13}$$

and the consumed tokens of draft model is, with (10) and (11),

$$T_2^k = \sum_{i=1}^{N} |\mathbf{q}_i| + \sum_{i=1}^{N} \sum_{k \in I_i^c} |l_k^i| + \sum_{i=1}^{N} \sum_{m=1}^{M} \sum_{k \in I_i^c} |l_{km}^i|. \tag{14}$$

## B.3 COMPARISON OF AVERAGE CONSUMPTION

Now we compare StepDPO with KEEP. Consider the difference of average of consumed tokens. We introduce the assumption that the expectation of $s_k^i$ or $l_k^i$ exhibits independence from the question and index of the step, that is $\mathbb{E}[s_k^i] = E[l_{km}^i] = \mathbb{E}[s]$. And in StepDPO, the expectation of the number of steps of an answer (no matter it is correct or not) is a constant, that is $\mathbb{E}[n_i] = \mathbb{E}[n_i'] = \mathbb{E}[n]$.

### B.3.1 AVERAGE CONSUMPTION OF THE TARGET MODEL

Thus, with (7), we have the average tokens consumption of target model in StepDPO:

$$\mathbb{E}_{s,n,n',\mathcal{I}}[\frac{T_1^s}{p_w p_r N}]$$

$$= \frac{1}{p_w p_r N}(\sum_{i=1}^{N}|\mathbf{q}_i| + \sum_{i=1}^{N}\mathbb{E}_{s_k^i,n_i}[\sum_{k=1}^{n_i}|s_k^i|] + p_w\sum_{i=1}^{N}(|\mathbf{q}_i| + \mathbb{E}_{\hat{s}_k^i,n_i'}[\sum_{k=1}^{n_i'}|\hat{s}_k^i|]))$$

$$= \frac{1}{p_w p_r N}(\sum_{i=1}^{N}|\mathbf{q}_i| + \sum_{i=1}^{N}\mathbb{E}_{n_i}[\sum_{k=1}^{n_i}\mathbb{E}[|s_k^i|]] + p_w\sum_{i=1}^{N}(|\mathbf{q}_i| + \mathbb{E}_{n_i'}[\sum_{k=1}^{n_i'}\mathbb{E}[|\hat{s}_k^i|]]))$$

$$= \frac{1+p_w}{p_w p_r N}\sum_{i=1}^{N}|\mathbf{q}_i| + \frac{1+p_w}{p_w p_r}\mathbb{E}[n]\mathbb{E}[s]. \tag{15}$$

In KEEP, experiments demonstrate that the number of key steps is typically half of the number of all steps, i.e., $\mathbb{E}[n_i^c] = \frac{1}{2}\mathbb{E}[n]$. With (13), we have:

$$\mathbb{E}_{l,n,n^c}[\frac{T_1^k}{p_a M \sum_{i=1}^{N} n_i^c}]$$

$$= \frac{2}{p_a N M \mathbb{E}[n]}((1 + \frac{1}{2}M\mathbb{E}[n])\sum_{i}^{N}|\mathbf{q}_i| + N\mathbb{E}[n]\mathbb{E}[s] + \frac{1}{2}MN\mathbb{E}[n]\mathbb{E}[s]),$$

where $n_i^c$ is the number of key steps of the answer of $\mathbf{q}_i$. In practice, $M$ is set 3, which means 3 different kinds of errors. Consequently,

$$\mathbb{E}_{l,n,n^c}[\frac{T_1^k}{p_a M \sum_{i=1}^{N} n_i^c}] = \frac{\frac{2}{3} + \mathbb{E}[n]}{p_a \mathbb{E}[n] N}\sum_{i}^{N}|\mathbf{q}_i| + \frac{5}{3 p_a \mathbb{E}[n]}\mathbb{E}[n]\mathbb{E}[s]. \tag{16}$$

Clearly, the comparative magnitude between (15) and (16) is contingent upon the parameters $p_r$, $p_w$, and $p_a$. Notice that $p_w$ is the probability of the target model giving an incorrect answer, while $p_r$ is that of giving a correct answer. In practice we find that for a easy dataset like GSM-8K, $p_r$ is approximately $40\%$ while $p_w$ is rather low——it takes average 20 times to collect an incorrect answer. The situation is opposite for a difficult dataset like MATH. In both situations we find that $p_w + p_r < 1$. However, in KEEP, since key steps have been selected, $p_a$ is rather high and usually over $50\%$, which is higher than both $p_r$ and $p_w$. Thus, we have

$$2p_w p_r < 2p_w(1 - p_w) < \frac{1}{2} < p_a.$$

Notice that a proper answer has at least 2 steps on multi-step mathematical reasoning tasks, which means $\mathbb{E}[n] \geq 2$. Thus, $\frac{3\mathbb{E}[n]}{5} > 1$. This leads to

$$\mathbb{E}_{s,n,n'}[\frac{T_1^s}{p_w p_r N}]$$

$$= \frac{1+p_w}{p_w p_r N}\sum_{i=1}^{N}|\mathbf{q}_i| + \frac{1+p_w}{p_w p_r}\mathbb{E}[n]\mathbb{E}[s]$$

$$> \frac{2}{p_a N}\sum_{i=1}^{N}|\mathbf{q}_i| + \frac{2}{p_a}\mathbb{E}[n]\mathbb{E}[s]$$

$$\geq \frac{3}{2}\frac{1}{p_a N} \cdot \frac{\mathbb{E}[n] + \frac{2}{3}}{\mathbb{E}[n]}\sum_{i=1}^{N}|\mathbf{q}_i| + \frac{2}{p_a}\mathbb{E}[n]\mathbb{E}[s]$$

$$> \frac{3}{2}\frac{1}{p_a N} \cdot \frac{\mathbb{E}[n] + \frac{2}{3}}{\mathbb{E}[n]}\sum_{i=1}^{N}|\mathbf{q}_i| + 2 \cdot \frac{5}{3 p_a \mathbb{E}[n]}\mathbb{E}[n]\mathbb{E}[s]$$

$$> \frac{3}{2}\mathbb{E}_{l,n,n^c}[\frac{T_1^k}{p_a M \sum_{i}^{N} n_i^c}].$$

Therefore, we show that the number of average tokens consumed of target model in KEEP is less than that in StepDPO. In addition, we can also see that the number of average tokens consumed of target model in StepDPO is at least $\frac{3}{2}$ **times greater** than that in KEEP. In practical multi-step mathematical reasoning scenarios, $\mathbb{E}[n]$ is often greater than 2, even up to tens of steps, which results in much more tokens consumed in StepDPO.

### B.3.2 AVERAGE CONSUMPTION OF THE TEACHER MODEL

Next we give the number of tokens consumed of the teacher model. For StepDPO, in practice, we find $\mathbb{E}[r_i] = \frac{1}{2}\mathbb{E}[n]$. Consequently,

$$
\mathbb{E}_{n,r,m,\mathcal{I}}\left[\frac{T_2^s}{p_w p_r N}\right] = \frac{1}{p_w p_r N}\left(p_w \sum_{i=1}^{N}|\mathbf{q}_i| + \sum_{i\in\mathcal{I}}\mathbb{E}_{n_i,s_k^i}\left[\sum_{k=1}^{n_i}|s_k^i|\right] + \sum_{i\in\mathcal{I}}\mathbb{E}_{r_i,m_k^i}\left[\sum_{k=1}^{r_i}m_k^i\right]\right)
$$

$$
= \frac{1}{p_r N}\sum_{i=1}^{N}|\mathbf{q}_i| + \frac{1}{p_r}\mathbb{E}[n]\mathbb{E}[s] + \frac{1}{2p_r}\mathbb{E}[n]\mathbb{E}[m],
$$

and for KEEP,

$$
\mathbb{E}_l\left[\frac{T_2^k}{p_a M \sum_{i=1}^{N} n_i^c}\right] = \frac{2}{p_a M N \mathbb{E}[n]}\left(\sum_i^{N}|\mathbf{q}_i| + \frac{1}{2}(M+1)N\mathbb{E}[n]\mathbb{E}[s]\right)
$$

$$
= \frac{2}{3 p_a N \mathbb{E}[n]}\sum_i^{N}|\mathbf{q}_i| + \frac{4}{3 p_a \mathbb{E}[n]}\mathbb{E}[n]\mathbb{E}[s].
$$

Since $\mathbb{E}[m] > \mathbb{E}[s]$, and $p_a > p_r$

$$
\frac{1}{p_r}\mathbb{E}[n]\mathbb{E}[s] + \frac{1}{2p_r}\mathbb{E}[n]\mathbb{E}[m] > \frac{3}{2p_r}\mathbb{E}[n]\mathbb{E}[s] > \frac{9\mathbb{E}[n]}{8}\frac{4}{3 p_a \mathbb{E}[n]}\mathbb{E}[n]\mathbb{E}[s].
$$

Thus,

$$
\mathbb{E}_{n,r,m,\mathcal{I}}\left[\frac{T_2^s}{p_w p_r \sum_i^{N} n_i^c}\right]
$$

$$
= \frac{3}{3 p_r N}\sum_{i=1}^{N}|\mathbf{q}_i| + \frac{1}{p_r}\mathbb{E}[n]\mathbb{E}[s] + \frac{1}{2p_r}\mathbb{E}[n]\mathbb{E}[m]
$$

$$
> \frac{3}{3 p_a N}\sum_i^{N}|\mathbf{q}_i| + \frac{1}{p_r}\mathbb{E}[n]\mathbb{E}[s] + \frac{1}{2p_r}\mathbb{E}[n]\mathbb{E}[m]
$$

$$
> \frac{3}{3 p_a N}\sum_i^{N}|\mathbf{q}_i| + \frac{9}{8}\mathbb{E}[n]\frac{4}{3 p_a \mathbb{E}[n]}\mathbb{E}[n]\mathbb{E}[s])
$$

$$
> \frac{9}{8}\mathbb{E}[n]\mathbb{E}_l\left[\frac{T_2^k}{p_a M \sum_{i=1}^{N} n_i^c}\right]
$$

$$
\geq \frac{9}{4}\mathbb{E}_l\left[\frac{T_2^k}{p_a M \sum_{i=1}^{N} n_i^c}\right],
$$

which shows that the number of average tokens consumed of the target model in KEEP is also less than that in StepDPO. Furthermore, we can also see that the number of average tokens consumed by the target model in StepDPO is at least $\frac{9}{4}$ **times greater** than in KEEP.

**The analysis above demonstrates that our framework achieves superior performance with lower computational cost.**

### B.4 STATISTICAL ANALYSIS OF SAMPLING RESPONSES

We evaluate the cost of sampling and uncontrollability of generation using a sampling-based method by setting the temperature of the LLM (Qwen2-7B-Instruct) to 1.0. We conduct 100 sampling

iterations for questions from the GSM8K and MATH datasets, recording the minimum number of samples required to generate correct and incorrect responses. The results are shown in Figures 2 and 3. For the GSM8K dataset, the average number of samples needed to generate a correct answer is 2.32, while incorrect answers require 33.33 samples. Notably, 1% of the questions fail to produce a correct answer after 100 samples, and 16% fail to produce an incorrect answer. In the MATH dataset, generating a correct answer requires an average of 21.52 samples, and an incorrect answer requires 6.75 samples, with 16% of questions failing to produce a correct answer. (Some methods also require sampling to generate correct answers)

These findings underscore the limitations of sampling-based methods for error exploration. Despite extensive sampling, certain errors remain elusive, suggesting that these approaches may overlook small-probability yet high-impact errors. In large-scale applications, even rare errors can accumulate, posing significant risks. The inherent uncontrollability and high cost of sampling make it inefficient for comprehensive error discovery, highlighting the importance of active error exploration.

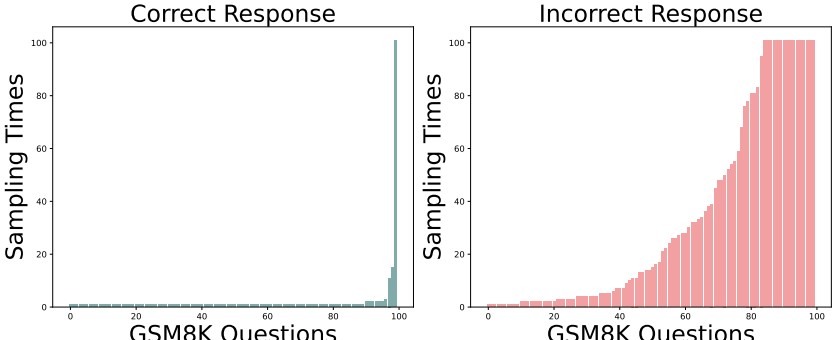

Figure 2: Statistics of sampling response based on GSM8K. The average sampling of correct answers is 2.32 times. The average sampling of incorrect answers is 33.33 times. 1% of questions cannot sample correct answers. 16% of questions cannot sample incorrect answers.

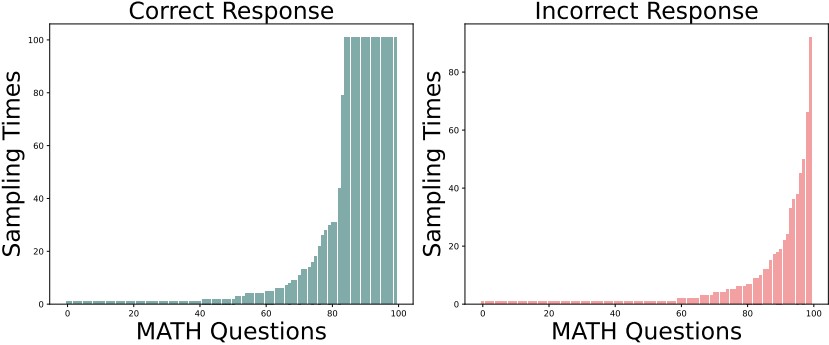

Figure 3: Statistics of sampling response based on MATH. The average sampling of correct answers is 21.52 times. The average sampling of incorrect answers is 6.75 times. 16% of questions cannot sample correct answers.

## B.5 EXPERIMENTAL RESULTS

We have conducted in-depth theoretical analysis and sampling experiments in the previous section. In order to more intuitively demonstrate the cost difference between our method (KEEP) and the sampling-based method (Step DPO), we conduct detailed experimental statistics based on the number of tokens input and output. As shown in Table 15. Results demonstrate that KEEP achieves significantly lower computational costs across all key processes. Specifically, Step DPO incurs high token costs primarily during incorrect answer sampling (4,329.08 tokens) and correct answer sampling (3,876.72 tokens). In contrast, KEEP's key step identification requires only 284.07 tokens,

Table 15: The results of computational costs based on the number of input and output tokens, assuming that one QA pair needs to generate one preference pair data, the following is the average results for one QA pair. But for KEEP, please note that our experiment assumes that we only need one keystep to generate one preference pair, so we need to divide the original cost by the number of keysteps to obtain the final results

| Step DPO | | | |
|---|---|---|---|
| Process | Model | Description | Token Costs in practice |
| Sampling of Incorrect Answers | target LLM | Input Q multiple times to sample A and obtain incorrect responses. | 4,329.08 |
| Error Step Localization | GPT - 4o | Input Q along with the incorrect A to output step - by - step analysis and the indices of error steps. | 566.94 |
| Sampling of Correct Answers | target LLM | Input Q along with the A up to the step before the error step multiple times, and sample to output the subsequent A to obtain the correct steps. | 3,876.72 |
| KEEP | | | |
| Process | Model | Description | Token Costs in practice |
| Key Step Identification | target LLM | Input Q and compute the PPL and Entropy of the output A (PPL and Entropy can be computed in one output). | 284.07 |
| Proactive Error Exploration | GPT - 4o | Input the QA pair along with the indices of key steps (selecting 50% of the steps as keysteps ), and output the 3 types error steps corresponding to the key steps. | 157.85 |
| Speculative Filtering | target LLM | Input Q and compute the output probabilities of the error steps in A. | 467.88 |

proactive error exploration uses 157.85 tokens, and speculative filtering consumes 467.88 tokens. Notably, **KEEP's total token costs represent approximately 1/10 of Step DPO on the target LLM**, validating its superior computational efficiency. This substantial reduction in token usage highlights KEEP's practical advantages for scalable implementation while maintaining data quality, as it minimizes reliance on expensive repeated sampling by focusing on critical step analysis and targeted error exploration.

## C  FURTHER ABLATION STUDY

### C.1  ABLATION ON MODEL ARCHITECTURE

To provide further insights, we conduct additional ablation experiments: SVPO + Speculative Filtering and SCDPO + Speculative Filtering, which involved adding our speculative filtering to the existing sampling based method for data construction. As shown in Table 16, the results demonstrate that existing methods generally achieve performance improvements on the GSM8K and MATH datasets when augmented with Speculative Filtering, with particularly significant gains observed on the MATH dataset. This highlights the effectiveness of our Speculative Filtering approach based on generation probabilities during the data construction phase. Moreover, these enhanced methods still underperform compared to KEEP, further underscoring the superiority of the KEEP framework.

Table 16: More results of ablation study on method combination

| Base model | Method | GSM8K | MATH |
|---|---|---|---|
| | SVPO | 86.7 | 52.2 |
| | SVPO+Speculative Filtering | 87.6 | 57.2 |
| Qwen2-7B-instruct | SCDPO | 86.8 | 51.6 |
| | SCDPO+Speculative Filtering | 87.3 | 55.3 |
| | **KEEP** | **89.1** | **59.5** |
| | SVPO | 84.8 | 48.7 |
| | SVPO+Speculative Filtering | 84.7 | 51.0 |
| Meta-Llama-3.1-8B-Instruct | SCDPO | 84.4 | 48.8 |
| | SCDPO+Speculative Filtering | 84.8 | 49.5 |
| | **KEEP** | **88.3** | **52.0** |

## D   THEORETICAL MOTIVATION FOR KEY-STEP IDENTIFICATION

In this section, we provide a theoretical justification for our PPL–Entropy scoring criterion. We start from a multiplicative approximation of final correctness, then connect step-wise uncertainty to expected log-likelihood improvement and conditional mutual information. This motivates our practical heuristic that combines step perplexity and entropy as a tractable proxy for information gain.

**Theorem D.1** (Step-level contribution to final correctness). *Let a problem instance $x$ with its gold step sequence $s_1, \ldots, s_N$ be given. Denote by $p_k = \pi_\theta(s_k \mid x, s_{1:k-1})$ the probability that the model $\pi_\theta$ generates the correct $k$-th step given context. Under the multiplicative approximation $P_{\mathrm{corr}}(\theta) \approx \prod_{k=1}^N p_k$, a small increment $\Delta p_j$ on step $j$ yields a change in log-correctness probability:*

$$\Delta \log P_{\mathrm{corr}} = \log(p_j + \Delta p_j) - \log p_j = \log\left(1 + \frac{\Delta p_j}{p_j}\right) \approx \frac{\Delta p_j}{p_j},$$

*where the last step holds for sufficiently small $\Delta p_j$. Thus, steps with smaller $p_j$ (i.e., higher perplexity / lower model confidence) yield larger potential log-likelihood improvement when corrected, highlighting them as natural optimization targets.*

*Proof.* The result follows from the definition of logarithm. For $\epsilon = \Delta p_j / p_j$, one has $\log(1 + \epsilon) = \epsilon - \epsilon^2/2 + \mathcal{O}(\epsilon^3)$. Therefore $\log(1 + \epsilon) \approx \epsilon$ for small $\epsilon$, proving the claim. $\square$

**Theorem D.2** (Information gain of a step and entropy upper bound). *Let $S_k$ be the random variable representing the token sequence at step $k$, and $Y \in \{0, 1\}$ the correctness of the final answer, conditioned on context $C = (x, s_{1:k-1})$. The conditional mutual information between $S_k$ and $Y$ is*

$$\mathrm{IG}_k := I(Y; S_k \mid C) = H(Y \mid C) - H(Y \mid S_k, C) = \mathbb{E}_{s_k \sim P(S_k \mid C)}\Big[\mathrm{KL}\big(P(Y \mid s_k, C) \,\|\, P(Y \mid C)\big)\Big].$$

*Moreover,*

$$\mathrm{IG}_k \leq \min\big(H(Y \mid C), H(S_k \mid C)\big).$$

*Thus, a step with higher conditional entropy $H(S_k \mid C)$ has a greater potential upper bound on information gain, suggesting stronger influence on final correctness (Cover, 1999).*

*Proof.* The equalities follow from the chain rule for mutual information and the KL divergence representation. The inequality follows from the standard property $I(A; B) \leq \min(H(A), H(B))$, applied to the conditional setting (Cover, 1999). $\square$

**Practical Scoring Criterion.**   Since computing $\mathrm{IG}_k$ directly is expensive, we use its entropy upper bound $H(S_k \mid C)$ as a tractable proxy. Our linear fusion $\alpha \cdot \mathrm{norm}(\mathrm{PPL}) + \beta \cdot \mathrm{norm}(\mathrm{Entropy})$ is a stable numerical relaxation of the multiplicative form suggested by the theory. This is consistent with prior uses of information gain in feature selection (Quinlan, 1986) and active learning (Settles, 2009), where entropy-based proxies have proven effective. Perplexity, as a classical measure of model uncertainty in language modeling (Jelinek, 1998), complements entropy by capturing step-level surprise.

**Monte-Carlo Verification Protocol.** We empirically validate the correlation between our proxy score and true expected improvement using the following Monte-Carlo estimation procedure (with $M = 5$ perturbations and $L = 10$ rollouts per step, random seeds fixed for reproducibility):

---

**Algorithm 1** Monte-Carlo estimation of step information gain

---

1: **for** each sample $(x, s_1, \ldots, s_N)$ **do**
2:    **for** each step $k$ **do**
3:       Compute $p_k = \pi_\theta(s_k \mid x, s_{1:k-1})$ and step entropy $H_k = \text{Entropy}(S_k \mid C)$.
4:       Compute proxy score $\text{Score}_k$ using our PPL-Entropy heuristic.
5:       Sample $M$ alternative realizations $\{s_k^{(m)}\}$ from a draft model given context $C$.
6:       **for** each $s_k^{(m)}$ **do**
7:          Estimate $P(Y = 1 \mid s_k^{(m)}, C)$ via $L$ rollouts to full solution.
8:       **end for**
9:       Estimate empirical distribution $P(Y \mid S_k, C)$ and compute $\text{IG}_k^{\text{MC}}$.
10:      Compute empirical expected improvement $\text{EI}_k^{\text{MC}} = (1/p_k) \cdot \text{IG}_k^{\text{MC}}$.
11:    **end for**
12: **end for**
13: Compute Spearman correlation between the proxy $\text{Score}_k$ and the empirical $\text{EI}_k^{\text{MC}}$.

---

**Verification Results.** To empirically validate our theoretical motivation, we conducted the Monte-Carlo estimation procedure outlined in Algorithm 1. The analysis was performed on 342 steps derived from 100 random samples of the GSM8K dataset. The results confirm a statistically significant positive correlation between our PPL–Entropy proxy score and the empirically estimated expected improvement (EI). We report a Spearman's rank correlation coefficient of $\rho = 0.488$ with a p-value of $7.79 \times 10^{-22}$, indicating a strong monotonic relationship. For completeness, Pearson correlation was $r = 0.314$ ($p < 10^{-8}$), which is weaker due to non-linear effects but still highly significant. This is appropriate as we care about rank-order for selecting key steps, not necessarily a linear fit. The scatter plot in Figure 4 visually corroborates this trend. This provides a principled, data-driven foundation for our key-step identification heuristic.

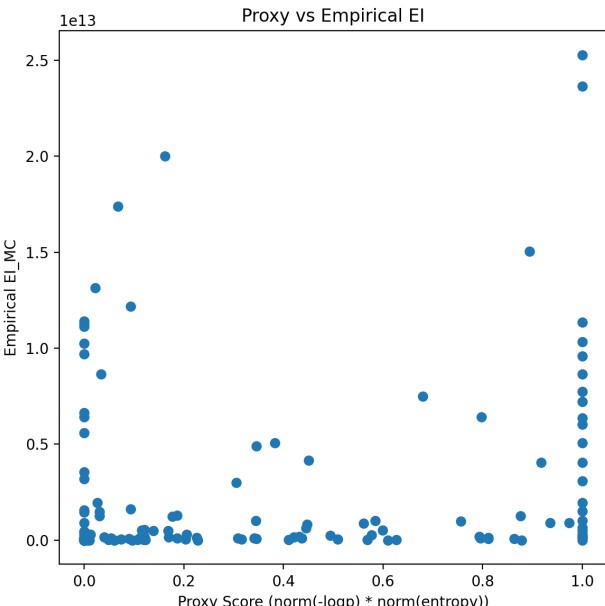

Figure 4: Monte Carlo validation of proxy score. Each point corresponds to a reasoning step. Higher proxy scores correlate with higher empirical error informativeness ($\text{EI}_{\text{MC}}$).

**Theoretical Assumptions and Practical Implementation** Our practical PPL-Entropy scoring rule is derived from our theoretical analysis by introducing several well-motivated assumptions to ensure computational tractability. We outline these key connections between theory and practice below:

- **Approximation of Correctness:** The multiplicative approximation in Theorem D.1 is a standard approach for modeling complex sequences, providing a tractable way to analyze step-wise contributions. While it simplifies the modeling of long-range dependencies, it effectively captures the local impact of step-level probabilities.

- **Entropy as an Information Gain Proxy:** Following established principles from information theory, we use conditional entropy as a computationally efficient proxy for information gain. Theorem D.2 confirms that entropy serves as a tight upper bound on the true information gain ($IG_k$), making it a standard and effective heuristic for identifying steps with high informational potential.

- **Principled Fusion of Metrics:** The linear fusion of PPL and entropy is a deliberate design choice to synthesize the insights from both theorems into a single, robust score. The theory validates the inclusion of both signals—model uncertainty (PPL) and information potential (entropy)—and our linear combination provides a simple yet effective method for balancing their contributions.

The empirical success of this design, validated by our Monte-Carlo verification, confirms that these principled approximations lead to a highly effective method for identifying high-impact reasoning steps for preference optimization.

## E    ANALYSIS OF PPL AND ENTROPY

To explore the role of Perplexity (PPL) and Entropy in identifying key steps, we conduct experiments on the GSM8k, MATH, and AQuA datasets. We randomly sample 1000 QA pairs and compute the PPL and Entropy for each step in the answers. We then calculate the percentage position of the steps with the highest PPL and Entropy in the answer (step index / total number of steps × 100%). As shown in Figure 5a, the results indicate that steps with higher PPL are concentrated in the latter parts of the answers, particularly between the 85% and 90% positions. This suggests that the model becomes more confused at critical steps near the conclusion. In contrast, Entropy is higher at the beginning, especially in the first sentence, likely due to its greater information content or prediction difficulty.

To more accurately identify key steps, we normalize and combine PPL and Entropy scores, then rank the results. As shown in Figure 5b, the combined Top 1 curve highlights the importance of both the beginning and the end. By selecting steps ranked in the Top 50% as key steps, we ensure a comprehensive coverage across different types of problems and step sequences. This analysis confirms the rationale and effectiveness of KEEP for identifying key steps.

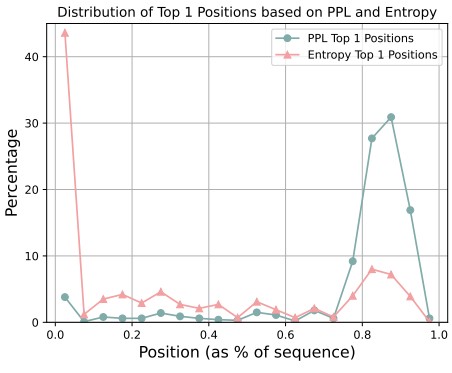  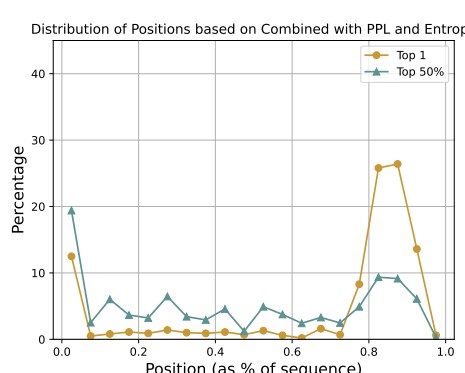

(a) Distribution of top 1 positions.    (b) Distribution of positions based on combined values.

Figure 5: Comparison of position distributions.

### E.1 MANUAL AND STATISTICAL ANALYSES

We conduct both manual and statistical analyses to provide further insights. As shown in Figure 6, Figure 7, and Figure 8, Entropy focuses on the importance of the current step within the context, while PPL emphasizes the complexity or importance of the step itself. The two have complementary characteristics. Therefore, we combine PPL and Entropy as indicators to identify key steps, defining them as either complex steps the model struggles with or core contextual steps. Further statistical analysis (see Figures 9 and 10 ) reveals that existing sampling methods tend to over-sample high-entropy steps while neglecting high-PPL steps. High-PPL steps may indicate higher complexity or model comprehension difficulties. Our method addresses this limitation by combining entropy and PPL to identify key steps.

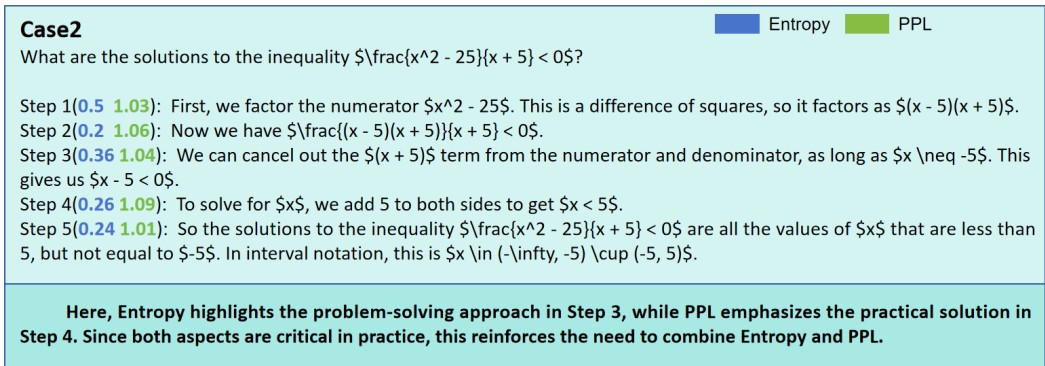

**Case 1**     Entropy    PPL

After the year 1994, during which year did sales increase by the greatest amount? Express your answer as a 4-digit year.

Step 1(**0.74 1.03**): We need to find out the increase in sales from one year to the next. To do this, we subtract the sales of the previous year from the sales of the current year.
Step 2(**0.87 1.04**): For 1995: $1,450,000 - $1,320,000 = $130,000$.
Step 3(**0.14 1.04**): For 1996: $1,710,000 - $1,450,000 = $260,000$.
Step 4(**0.14 1.04**): For 1997: $2,000,000 - $1,710,000 = $290,000$.
Step 5(**0.12 1.03**): For 1998: $2,300,000 - $2,000,000 = $300,000$.
Step 6(**0.99 1.12**): Comparing the increases, we see that the increase in sales is the greatest for 1998.

**In this example, Entropy focuses on high-information steps in context, often the first sentence with changes (e.g., Step 2 and the final step). PPL, however, assesses steps based solely on their inherent complexity, ignoring context (e.g., similar PPL values for Steps 2–5). Thus, both Entropy and PPL should be considered together.**

Figure 6: Case Study for Entropy and PPL

**Case2**     Entropy    PPL

What are the solutions to the inequality $\frac{x^2 - 25}{x + 5} < 0$?

Step 1(**0.5 1.03**): First, we factor the numerator $x^2 - 25$. This is a difference of squares, so it factors as $(x - 5)(x + 5)$.
Step 2(**0.2 1.06**): Now we have $\frac{(x - 5)(x + 5)}{x + 5} < 0$.
Step 3(**0.36 1.04**): We can cancel out the $(x + 5)$ term from the numerator and denominator, as long as $x \neq -5$. This gives us $x - 5 < 0$.
Step 4(**0.26 1.09**): To solve for $x$, we add 5 to both sides to get $x < 5$.
Step 5(**0.24 1.01**): So the solutions to the inequality $\frac{x^2 - 25}{x + 5} < 0$ are all the values of $x$ that are less than 5, but not equal to $-5$. In interval notation, this is $x \in (-\infty, -5) \cup (-5, 5)$.

**Here, Entropy highlights the problem-solving approach in Step 3, while PPL emphasizes the practical solution in Step 4. Since both aspects are critical in practice, this reinforces the need to combine Entropy and PPL.**

Figure 7: Case Study for Entropy and PPL

We include the changes in the model's average log probabilities for positive and negative examples (using 1000 responses generated by the initial model) during training. As shown in Figure 11, the gap in log probabilities between positive and negative examples widens during training, increasing from 0.59 to 1.03. This indicates that the model's ability to distinguish between examples improves significantly as training progresses.

## F QUALITY ANALYSIS OF GENERATED PREFERENCE DATA

To provide a deep validation of the data quality produced by our pipeline, we conducted a series of quantitative and qualitative analyses. These analyses confirm that our Speculative Filtering is effective and that the generated errors are both high-purity and pedagogically valuable.

**Case 3**

Entropy    PPL

Identify the values of $x$ for which the inequality $2x^2 + 8x$ is at most $-6$ and express your answer as an interval.

Step 1:(**0.3 1.07**):  First, we rewrite the inequality as $2x^2 + 8x + 6 \le 0$.
Step 2:(**0.5 1.04**):  Next, we factor out the leading coefficient of $2$ from the left-hand side of the inequality: $2(x^2 + 4x + 3) \le 0$.
Step 3:(**0.22 1.08**):  We then factor the quadratic expression inside the parentheses: $2(x + 1)(x + 3) \le 0$.
Step 4:(**0.25 1.01**):  Now, we find the critical points by setting each factor equal to zero: $x + 1 = 0$ gives $x = -1$, and $x + 3 = 0$ gives $x = -3$. These are the points where the expression changes its sign.
Step 5:(**0.98 1.19**):  We now consider the sign of each factor in the intervals determined by the critical points:
Step 6:(**0.13 1.04**):  - For $x < -3$, both $(x + 1)$ and $(x + 3)$ are negative, so their product is positive.
Step 7:(**0.03 1.04**):  - For $-3 < x < -1$, $(x + 1)$ is negative and $(x + 3)$ is positive, so their product is negative.
Step 8:(**0.18 1.04**) :  - For $x > -1$, both $(x + 1)$ and $(x + 3)$ are positive, so their product is positive.
Step 9:(**0.51 1.04**):  Since we want the expression to be at most zero, we are interested in the interval where the expression is negative. That is, the solution to the inequality is the interval $-3 \le x \le -1$.
Step 10:(**0.4 1.01**):  Therefore, the values of $x$ for which the inequality $2x^2 + 8x \le -6$ is true are $x \in [-3, -1]$.

**In this case, both Entropy and PPL identify Step 5 as pivotal, as it unlocks subsequent steps. This shows that despite their different evaluation angles, both metrics effectively capture key steps, underscoring their respective validity.**

Figure 8: Case Study for Entropy and PPL

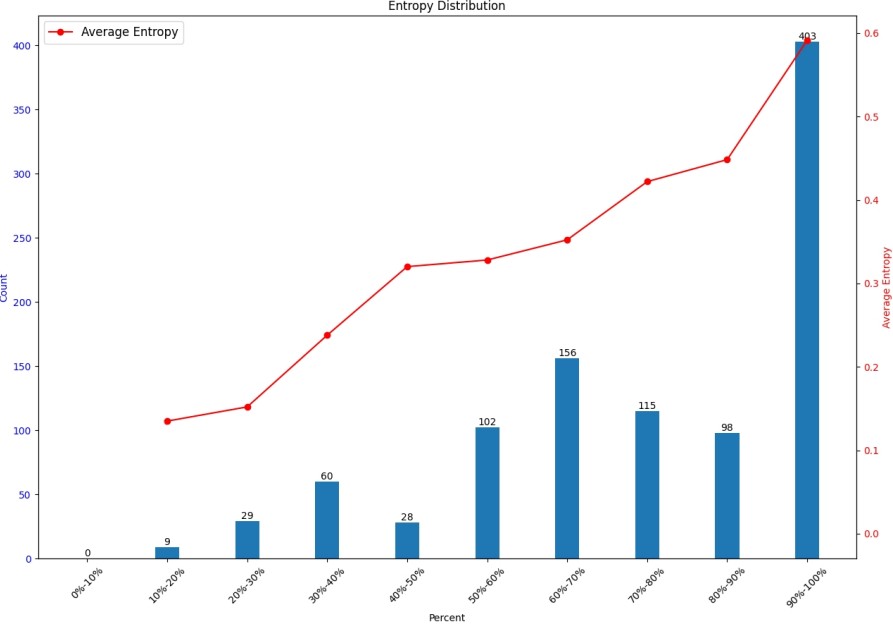

Figure 9: We analyze the data generated by the StepDPO method by randomly sampling 1,000 data points. The x-axis of the graph represents the index of the sampled error steps within the entire response, where 0-10% indicates the first 10% of the response, and 90-100% indicates the later parts. The y-axis represents the number of samples and the average Entropy. From the figure, it can be observed that nearly half of the samples are located in the later parts of the response and have relatively high Entropy values.

## F.1 Effectiveness of Speculative Filtering

To quantitatively measure the impact of our Speculative Filtering module, we analyzed the generation probabilities of preference pairs before and after filtering. The core motivation of this module is to mitigate distribution shift from the external draft model by retaining "hard" negative samples that are

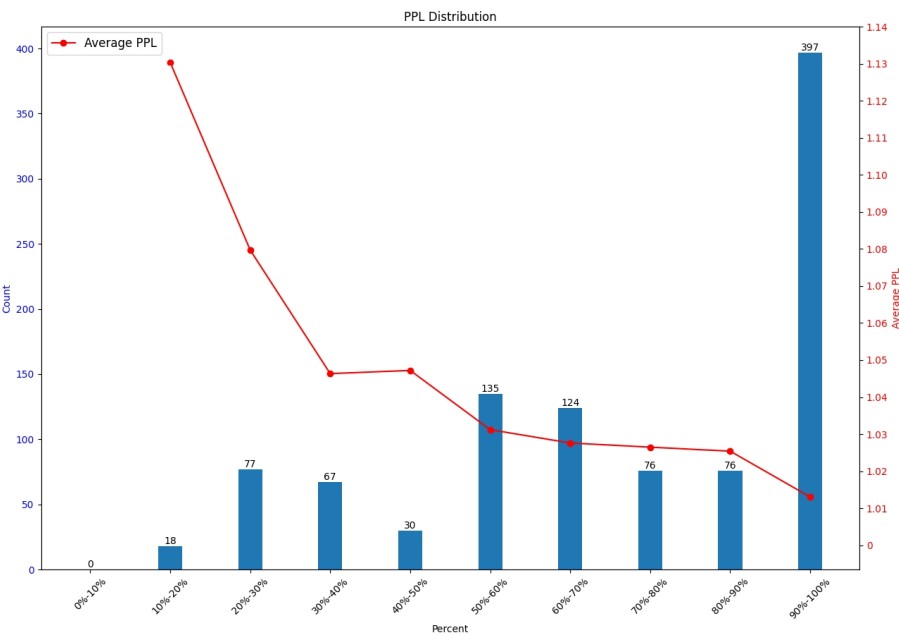

Figure 10: We analyze the data generated by the StepDPO method by randomly sampling 1,000 data points. The x-axis of the graph represents the index of the sampled error steps within the entire response, where 0-10% indicates the first 10% of the response, and 90-100% indicates the later parts. The y-axis represents the number of samples and the average PPL. From the figure, it can be observed that nearly half of the samples are located in the later parts of the response and have relatively low PPL values.

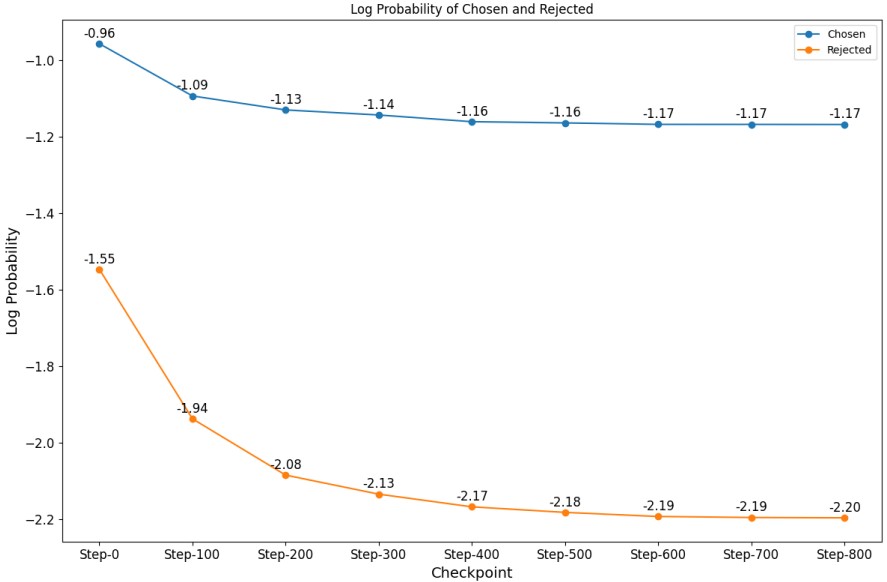

Figure 11: The gap in log probabilities between positive and negative examples

closer to the target model's ability frontier. As shown in Table 17, the module successfully reduces the probability ratio between chosen and rejected steps from 1.95 to 1.34. This confirms that we

are selecting more challenging and relevant negative examples, ensuring a more stable and effective preference tuning process.

Table 17: Quantitative impact of the Speculative Filtering module.

| Metric (Mean) | Before Filtering | After Filtering |
|---|---|---|
| Chosen Step Probability | 0.7273 | 0.7149 |
| Rejected Step Probability | 0.4281 | 0.5423 |
| **Ratio (Chosen / Rejected)** | **1.9578** | **1.3432** |

### F.2 GRADIENT-BASED ATTRIBUTION OF ERROR IMPACT

To analyze which types of errors are most beneficial for preference learning, we conducted a detailed gradient-based attribution analysis. We annotated 1,000 samples with fine-grained error types and computed the gradient norm and cosine similarity during KEEP training to estimate each type's contribution to learning. As shown in Table 18, semantically meaningful errors (e.g., misunderstanding conditions, counting errors) lead to stronger learning signals (higher gradient norm and cosine similarity), while surface-level errors (e.g., substitution mistakes) contribute less. This confirms that KEEP's proactive exploration process naturally generates pedagogically high-impact training signals.

Table 18: Gradient-based attribution analysis of error types.

| Error Type | #Samples | ↑ Grad Norm | ↑ Cosine Sim. |
|---|---|---|---|
| Multiplication errors | 44 | 0.0575 | 0.0151 |
| Misunderstanding the question | 208 | 0.0529 | 0.0139 |
| Counting errors | 76 | 0.0505 | 0.0132 |
| Improper division | 47 | 0.0457 | 0.0120 |
| Addition errors | 76 | 0.0452 | 0.0118 |
| Decimal calculation errors | 109 | 0.0444 | 0.0116 |
| Subtraction errors | 34 | 0.0443 | 0.0116 |
| Misunderstanding variable relations | 246 | 0.0427 | 0.0112 |
| Fraction operation errors | 46 | 0.0385 | 0.0101 |
| Substitution mistakes | 103 | 0.0310 | 0.0081 |
| Other | 11 | 0.0062 | 0.0016 |

### F.3 HUMAN AUDIT OF PREFERENCE PAIR QUALITY

To quantify the quality of the final dataset after speculative filtering, we conducted a manual audit on 1,000 randomly sampled preference pairs. Our analysis, summarized in Table 19, confirms the data is of high purity, with a total potential noise level under 5%. The minor noise primarily stems from occasional inaccurate step segmentation or instances where a rejected step represents an alternative valid solution path. This audit validates that our automated pipeline reliably produces a high-quality dataset of challenging and informative preference pairs suitable for robust training.

## G ANALYSIS OF ERROR TYPES

Using GPT-4o, We analyze the error types in equal amounts of data generated by different methods. As shown in Figures 12, 13, and 14, existing sampling methods like Step DPO exhibit a clear long-tail distribution in error types, with most errors concentrated in a few categories. In contrast, KEEP effectively mitigated this issue and generated more diverse error types, avoiding the "mode collapse" issue and providing a richer training signal for the model.

Table 19: Analysis of potential noise in post-filter error step (1,000-sample audit).

| Noise Type in Error Step | Description | Percentage |
|---|---|---|
| Inaccurate Step Segmentation | The content in 'Chosen' and 'Rejected' steps is too short or interrupted. | 2.7% |
| False Generation Error | The 'rejected' step was not factually wrong but represented a different, valid solution path. | 2.1% |
| **Total Potential Noise** | **The final dataset is confirmed to be of high purity.** | **< 5.0%** |

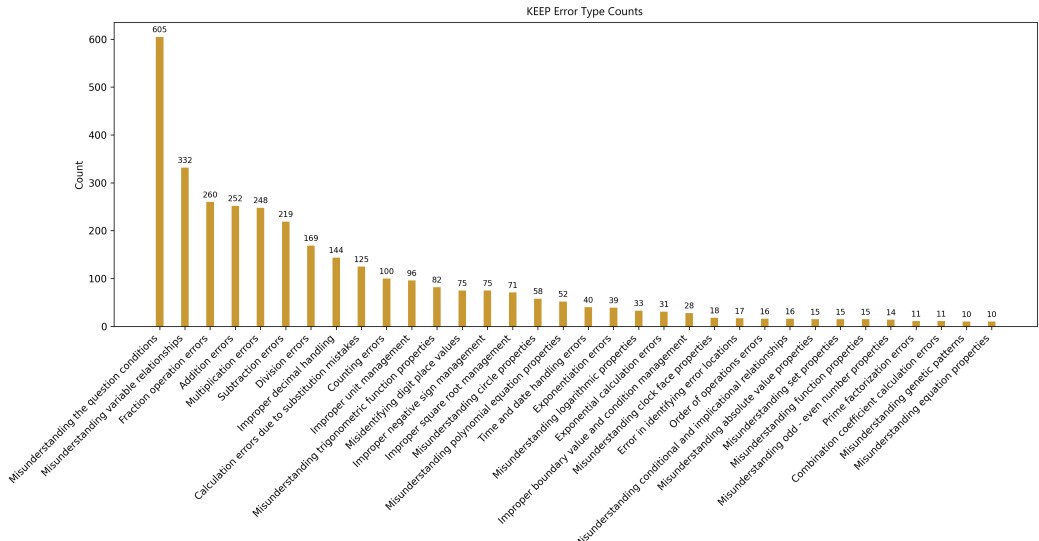

Figure 12: Analysis of error types based on KEEP

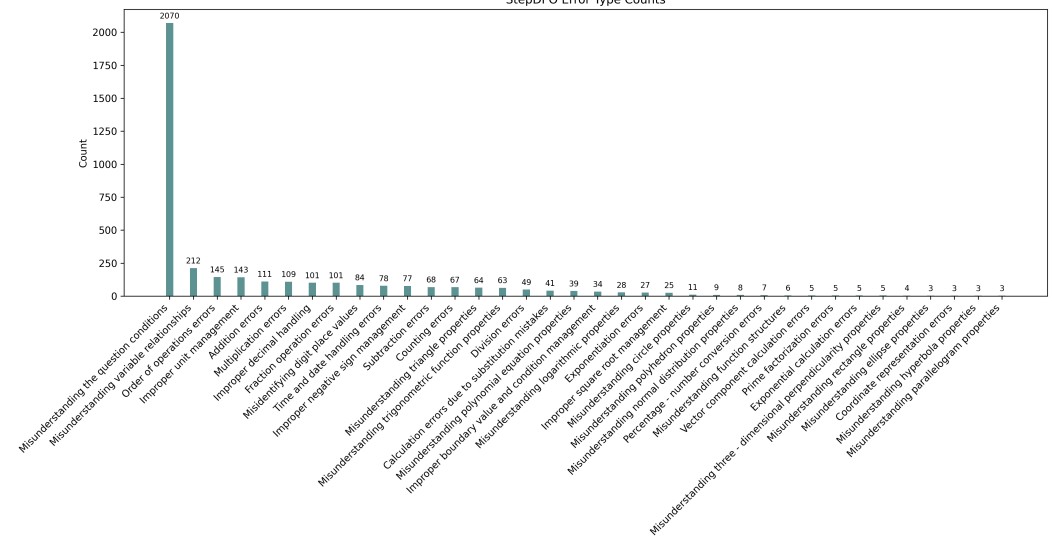

Figure 13: Analysis of error types based on StepDPO

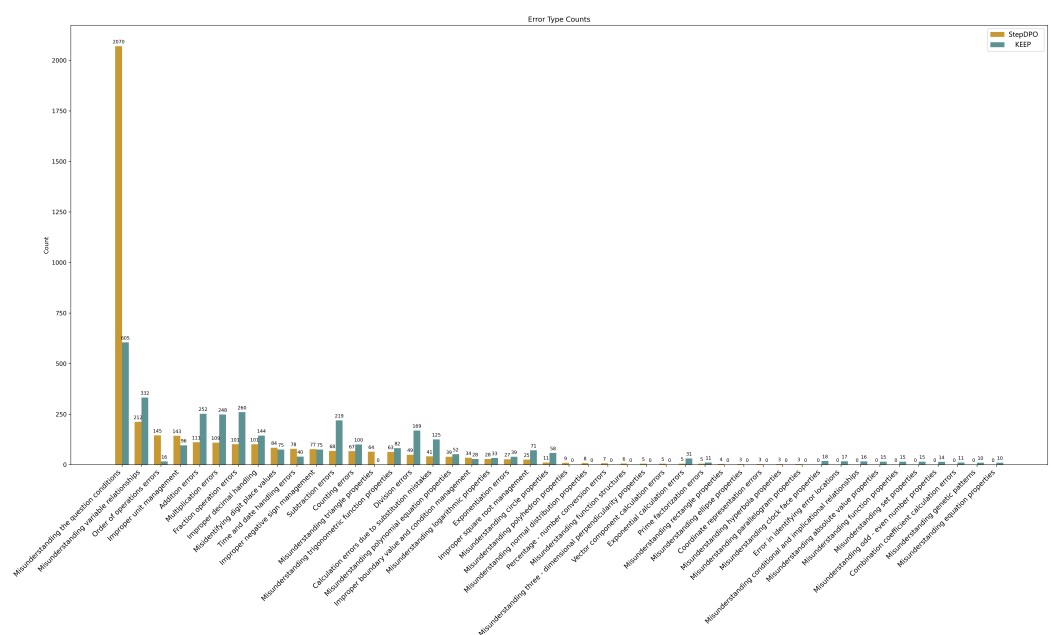

Figure 14: Analysis of error types based on KEEP and StepDPO

## H  CASE STUDY

### H.1  CASE STUDY ON QWEN2-72B SERIES MODELS

Respectively, Table 20 and Table 21 present the different generation solutions of KEEP-QWEN2-72B and Qwen2-72B-Instruct for the same problem from AIME24. we observe that the solutions generated by Qwen2-72B-Instruct contain an error. Specifically, Qwen2-72B-Instruct ignored the condition that the list is composed of integers after calculating the median and adding it to the list. KEEP-QWEN2-72B, however, did not make such errors.

### H.2  CASE STUDY ON LLAMA-3.1-70B SERIES MODELS

Respectively, Table 22 and Table 23 present the different generation solutions of KEEP-LLAMA-3.1-70B and Llama-3.1-70B-Instruct for the same problem from Odyssey-MATH. we observe that the solutions generated by Llama-3.1-70B-Instruct contain an error. Specifically, Llama-3.1-70B-Instruct incorrectly assumes that the given side is the hypotenuse, while in fact, it is the longer leg. KEEP-LLAMA-3.1-70B, however, did not make such errors.

## I  USE OF LARGE LANGUAGE MODELS IN MANUSCRIPT PREPARATION

In accordance with ICLR 2026 policy, we disclose the use of Large Language Models (LLMs) as an assistive tool in the preparation of this manuscript. The primary application of LLMs was to aid in improving the clarity and quality of the writing.

Our process involved using an LLM to perform the following specific tasks:

- **Grammar and Spelling Correction:** Identifying and correcting grammatical errors and spelling mistakes.
- **Clarity and Readability Enhancement:** Rephrasing sentences and suggesting alternative phrasings to improve the overall readability and flow of the text.
- **Conciseness:** Assisting in shortening sentences and paragraphs to make the writing more direct and concise.

The core scientific contributions, analyses, and claims presented in this paper are the work of the human authors. We have ensured that the use of LLMs in the writing process was conducted responsibly and in line with academic and ethical standards.

Table 20: An example of the generated solution by KEEP-QWEN2-72B.

**Problem 1**:
A list of positive integers has the following properties:

- The sum of the items in the list is 30.

- The unique mode of the list is 9.

- The median of the list is a positive integer that does not appear in the list itself.

Find the sum of the squares of all the items in the list.
**Reference Answer**:
236

**KEEP-QWEN2-72B** :
To solve this problem, we need to construct a list of positive integers that satisfies the given conditions and then find the sum of the squares of all the items in the list.
Given conditions:

1. The sum of the items in the list is 30.

2. The unique mode of the list is 9.

3. The median of the list is a positive integer that does not appear in the list itself.

Let's start by addressing the mode condition. Since 9 is the unique mode, it must appear more frequently than any other number in the list. To ensure that 9 is the unique mode, we can start by including 9 twice in the list. This gives us a sum of 18 so far ($9 + 9 = 18$).
Next, we need to ensure that the sum of the list is 30. This means we need to add numbers to the list that sum to 12 ($30 - 18 = 12$). To keep 9 as the unique mode, we cannot add any more 9s to the list. We also need to ensure that the median is a positive integer that does not appear in the list itself.
Let's consider the median condition. Since the list must have an odd number of elements for there to be a single median, and we already have 2 elements (9 and 9), we need to add at least 3 more elements to the list to make the total number of elements odd. To keep the sum at 30, we can add 1, 3, and 8 to the list. This gives us a list of 9, 9, 1, 3, and 8, which sums to 30.
The median of this list is 8, which does not appear in the list itself, satisfying the third condition. The list is now 9, 9, 1, 3, and 8.
Finally, we need to find the sum of the squares of all the items in the list. This is calculated as follows:

$$9^2 + 9^2 + 1^2 + 3^2 + 8^2 = 81 + 81 + 1 + 9 + 64 = 236$$

Therefore, the sum of the squares of all the items in the list is $\boxed{236}$.

Table 21: An example of the generated solution by the general Qwen2-72B-Instruct. The red box highlights the error.

---

**Problem 1**:
A list of positive integers has the following properties:

- The sum of the items in the list is 30.
- The unique mode of the list is 9.
- The median of the list is a positive integer that does not appear in the list itself.

**Reference Answer**:
236

---

**Qwen2-72B-Instruct**:
To solve this problem, we need to construct a list of positive integers that satisfies the given conditions and then find the sum of the squares of all the items in the list.
Given conditions:

1. The sum of the items in the list is 30.
2. The unique mode of the list is 9.
3. The median of the list is a positive integer that does not appear in the list itself.

Let's start by addressing the mode condition. Since 9 is the unique mode, it must appear more frequently than any other number in the list. To ensure that 9 is the unique mode, we can start by including 9 twice in the list. This gives us a sum of 18 so far.
Next, we need to ensure that the sum of the list is 30. This means we need to add numbers that sum up to 12 to the list. We also need to ensure that none of these numbers appear more frequently than 9 to maintain 9 as the unique mode. To achieve this, we can add the numbers 1, 2, and 3 to the list, which adds up to 6. This gives us a total sum of 24 so far. To reach 30, we need to add another number, which can be 6.
So, the list so far is 9, 9, 1, 2, 3, 6. This list has a sum of 30 and 9 is the unique mode.
Now, let's address the median condition. The median of the list is the middle number when the list is arranged in ascending order. Since we have 6 numbers, the median will be the average of the 3rd and 4th numbers. To ensure that the median does not appear in the list, we can re-arrange the list to be 1, 2, 3, 6, 9, 9. The median is then the average of 3 and 6, which is 4.5.
Since 4.5 is not an integer, it cannot appear in the list, satisfying the third condition.
Finally, we need to find the sum of the squares of all the items in the list. The list is 1, 2, 3, 6, 9, 9. The sum of the squares is

$$1^2 + 2^2 + 3^2 + 6^2 + 9^2 + 9^2 = 1 + 4 + 9 + 36 + 81 + 81 = 212$$

Therefore, the sum of the squares of all the items in the list is $\boxed{212}$.

---

Table 22: An example of the generated solution by KEEP-LLAMA-3.1-70B.

**Problem 2**:
In $\triangle$ABC, AB $= 10$ cm, $\angle$B $= 90°$, and $\angle$C $= 60°$. Determine the length of BC.

A) $10$ cm
B) $10\sqrt{3}$ cm
C) $\frac{10\sqrt{3}}{3}$ cm
D) $20$ cm

**Reference Answer**:
$C$

**KEEP-LLAMA-3.1-70B**:
To find the length of $BC$ in $\triangle ABC$, we can use trigonometric ratios since we have information about the angles and one side.
**Given:**

- $AB = 10$ cm
- $\angle B = 90°$ (right angle)
- $\angle C = 60°$

Since $\angle B$ is $90°$, $\triangle ABC$ is a right-angled triangle. We can use the trigonometric ratios for a 30-60-90 triangle, which are in the ratio $1 : \sqrt{3} : 2$ for the sides opposite the angles.
In this case, $\angle C$ is $60°$, so the side opposite to it ($AB$) will be the side multiplied by $\sqrt{3}$ in the ratio. However, we are given $AB$, and we need to find $BC$, which is the side opposite the $30°$ angle. Since the sum of angles in a triangle is $180°$, we have:

$$90° + 60° = 150°,$$

and thus the remaining angle $A$ must be $30°$.
In a 30-60-90 triangle, the side opposite the $30°$ angle ($BC$ in this case) is half the hypotenuse ($AC$) or the side opposite the $60°$ angle ($AB$) divided by $\sqrt{3}$.
However, we don't have the hypotenuse ($AC$) directly, but we can calculate $BC$ using $AB$ and the properties of a 30-60-90 triangle. Since $AB$ is the side opposite the $60°$ angle, it's the side that would be multiplied by $\sqrt{3}$ in the ratio. To find $BC$, we need to divide $AB$ by $\sqrt{3}$ to get the side opposite the $30°$ angle:

$$BC = \frac{AB}{\sqrt{3}}$$

$$BC = \frac{10}{\sqrt{3}}$$

To rationalize the denominator, we multiply the numerator and denominator by $\sqrt{3}$:

$$BC = \frac{10 \times \sqrt{3}}{\sqrt{3} \times \sqrt{3}}$$

$$BC = \frac{10\sqrt{3}}{3}$$

Thus, the final answer is:

$$\boxed{C}$$

Table 23: An example of the generated solution by the general Llama-3.1-70B-Instruct. The red box highlights the error.

**Problem 2**:
In △ABC, AB = 10 cm, ∠B = 90°, and ∠C = 60°. Determine the length of BC.

$$\begin{aligned}
&\text{A)} \quad 10 \, \text{cm} \\
&\text{B)} \quad 10\sqrt{3} \, \text{cm} \\
&\text{C)} \quad \tfrac{10\sqrt{3}}{3} \, \text{cm} \\
&\text{D)} \quad 20 \, \text{cm}
\end{aligned}$$

**Reference Answer**:
$C$

**Llama-3.1-70B-Instruct**:
To find the length of $BC$ in $\triangle ABC$, we can use trigonometric ratios since we have information about the angles and one side.
**Given:**

- $AB = 10$ cm

- $\angle B = 90°$ (right angle)

- $\angle C = 60°$

Since $\angle B$ is $90°$, $\triangle ABC$ is a right-angled triangle. We can use the trigonometric ratios for a 30-60-90 triangle, which are in the ratio $1 : \sqrt{3} : 2$ for the sides opposite the angles.
In this case, $\angle C$ is $60°$, so the side opposite to it ($AB$) will be the side multiplied by $\sqrt{3}$ in the ratio, and the side adjacent to it ($BC$) will be the side multiplied by 1 in the ratio.
However, we are given $AB$, which is the side opposite the $30°$ angle (not directly given, but since it's a right triangle with a $60°$ angle, the other acute angle must be $30°$). $AB$ corresponds to the side multiplied by 1 in the ratio (for the $30°$ angle), but we need to find $BC$, which corresponds to the side multiplied by $\sqrt{3}$ in the ratio for the $60°$ angle, but in relation to the side opposite the $30°$ angle. So, if $AB = 10$ cm, and considering the 30-60-90 triangle ratio, $AB$ would be the "1" side, and $BC$ would be the "$\sqrt{3}$" side. However, since $AB$ is given as 10 cm, which aligns with the "2" side in a standard 30-60-90 triangle (where the ratio is $1 : \sqrt{3} : 2$), we need to adjust our calculation.
In a standard 30-60-90 triangle, if the "2" side (hypotenuse) is 10 cm , the "1" side (opposite the $30°$ angle) would be $10/2 = 5$ cm, and the "$\sqrt{3}$" side (opposite the $60°$ angle) would be $5\sqrt{3}$ cm.
However, we are looking for $BC$, which is opposite the $60°$ angle, and we've established that $AB$ (the side we have) is actually the "2" side in the ratio due to its length and the given angles. So, we need to find the length of $BC$ based on $AB$ being the hypotenuse of the triangle, which means we actually look at $AB$ as the "2" side in the ratio, and calculate $BC$ as the "$\sqrt{3}$" side.
Given that $AB = 10$ cm, and it's the "2" side, to find the "1" side (which would help us scale the "$\sqrt{3}$" side correctly), we divide $AB$ by 2 to get the base "1" side, which is 5 cm. Then, we multiply this by $\sqrt{3}$ to find $BC$.

$$BC = 5 \times \sqrt{3} = \frac{10}{2} \times \sqrt{3} = 5\sqrt{3} \text{ cm.}$$

However, simplifying this with the given options, we see that the correct representation of our calculation is actually:

$$BC = \frac{10\sqrt{3}}{2} = 5\sqrt{3} \text{ cm.}$$

Thus, the final answer is:

$$\boxed{\frac{10\sqrt{3}}{2} = 5\sqrt{3}}$$

