# OpenReview forum: "Preference Optimization via Key-step Error Exploration for Multi-step Reasoning in LLMs"
_ICLR.cc/2026/Conference — ICLR 2026 Conference Desk Rejected Submission_

### Official Review · Reviewer_kwVS · 2025-10-30

**Soundness:** 3
**Presentation:** 3
**Contribution:** 3
**Rating:** 6
**Confidence:** 4

**Summary:**

The paper introduces a method for preference optimization through key-step error exploration, which comprises three stages: key step identification, proactive error exploration, and speculative filtering. Specifically, the paper employs PPL and entropy to identify key steps in the long-chain reasoning trajectories. It then proactively explores potential errors using GPT-4o, and applies speculative filtering based on rejection sampling to retain high-valued data. The resulting fine-grained preference dataset is subsequently used to train the model with step-dpo.

The paper conducts experiments on six mathematical reasoning benchmarks and other general reasoning benchmarks to evaluate the effectiveness and generality of the method.

**Strengths:**

The paper is well written and easy to follow.

The paper combines PPL and information entropy to identify key steps, and also has a theoretical analysis.

The experimental results demonstrate the effectiveness of the proposed method.

**Weaknesses:**

The paper combines PPL and information entropy to identify key steps. Although the ablation study confirms the importance of key step identification, it does not clarify the respective contributions of these two factors.

In the proactive error exploration stage, although the paper generates incorrect steps at three different levels, it does not specify their respective impact in the experiments. Furthermore, the training process lacks clarification on how these data are utilized, nor does it explain how the proportions of these data are allocated in the experiments.

As shown in Tables 2 and 3, the proposed method KEEP appears to exhibit similar or even worse performance than RISE. It would be helpful to provide an explanation for these observations.

Additionally, the experiments in this paper are based on Llama-3.1-8B and Qwen2-7B as backbones. To more rigorously evaluate the method, it would be beneficial to adopt more advanced backbones such as Qwen3-8B and DeepSeek-R1-Distill-Qwen-7B, which already surpass the performance achieved by the KEEP in the paper.

Several relevant studies on DPO, particularly step-wise DPO, such as IUPO[1], Selective-DPO [2], ConfPO[3] and Full-Step-DPO[4], have been proposed this year. However, the paper fails to incorporate these works in the related work section.

[1] Uncertainty-Aware Iterative Preference Optimization for Enhanced LLM Reasoning, ACL 2025 \
[2] Not All Preferences are What You Need for Post-Training: Selective Alignment Strategy for Preference Optimization, 2025.7 \
[3] ConfPO: Exploiting Policy Model Confidence for Critical Token Selection in Preference Optimization, 2025.6 \
[4] Full-Step-DPO: Self-Supervised Preference Optimization with Step-wise Rewards for Mathematical Reasoning, 2025.2

**Questions:**

see the weaknesses.

---

> ### Author Response · Authors · 2025-11-25
> **Response to Reviewer kwVS (part 1)**
>
> **We thank the reviewer for the insightful and valuable comments. We respond to each comment as follows and sincerely hope that our rebuttal could properly address your concerns. If so, we would deeply appreciate it if you could raise your score. If not, please let us know your further concerns, and we will continue actively responding to your comments and improving our submission.**
>
> ---
>
> # **Weakness 1 — Response: Respective Contributions of PPL and Entropy**
>
> We thank the reviewer for this insightful comment. We agree that clarifying the distinct roles of Perplexity (PPL) and Entropy is crucial. While both relate to uncertainty, our analysis reveals they capture **orthogonal dimensions** of reasoning difficulty.
>
> We clarify their **respective contributions** from three perspectives: **(1) Theoretical Role**, **(2) Empirical Mechanism**, and **(3) Quantitative Contribution**.
>
> ## **(1) Theoretical Role: Magnitude vs. Uncertainty**
>
> In **Appendix D**, we derive the theoretical foundation for why these two specific metrics are chosen as a tractable proxy for Information Gain (IG).
>
> * **PPL (The Magnitude of Improvement):** Related to **Theorem D.1**, PPL measures the model's inverse likelihood of generating the step. Theoretically, steps with high PPL (low probability) offer the mathematically largest potential for log-likelihood improvement if corrected. **Its contribution is identifying *where* the model is weak.**
> * **Entropy (The Information Bound):** Related to **Theorem D.2**, conditional entropy serves as an upper bound for the **Mutual Information** between the current step and the final outcome. **Its contribution is identifying *which* steps are structurally critical to the reasoning path.**
>
> **Synthesis:** The linear fusion effectively approximates the **local KL divergence gap**, pinpointing steps that are both *hard to execute* (High PPL) and *logically influential* (High Entropy).
>
> ## **(2) Empirical Mechanism: Inherent Complexity vs. Contextual Uncertainty**
>
> We distinguish their mechanisms based on the detailed Case Studies in **Appendix E (Figures 6-8)**. As shown in the qualitative analysis, they target fundamentally different reasoning failures:
>
> * **PPL (Inherent Complexity):**
>     * *Definition:* Reflects the model's unfamiliarity with the specific content of a step, largely independent of context.
>     * *Case Evidence:* As shown in **Figure 6** and **Figure 7**, PPL assesses steps based on "inherent complexity" , often highlighting practical solution steps or calculation bottlenecks.
>     * *Contribution:* It captures **"Execution Failures"**—where the model knows *what* to do but struggles *how* to do it correctly (e.g., arithmetic errors).
>
> * **Entropy (Contextual Uncertainty):**
>     * *Definition:* Reflects the uncertainty of the next token distribution *given the context*, focusing on information content.
>     * *Case Evidence:* As shown in **Figure 6**, Entropy focuses on "high-information steps in context". Similarly, in **Figure 7**, Entropy highlights the "problem-solving approach", marking critical logical transitions.
>     * *Contribution:* It captures **"Logic Failures"**—where the model faces a branching point and is uncertain about the direction (e.g., choosing a solving strategy).
>
> **Insight:** Using only one metric leads to blind spots. As noted in **Figure 6**, PPL may ignore context-dependent shifts that Entropy captures, while Entropy may miss calculation-heavy steps that PPL identifies. **Fusion is necessary to capture the full spectrum of reasoning failure modes.**
>
> ## **(3) Quantitative Contribution: Ablation Study**
>
> To quantify their respective impacts, we conducted an ablation study on **Qwen2-7B** and **Llama-3.1-8B**.
>
> **Table. Ablation of Individual Metrics vs. Fusion (Accuracy %)**
>
> |Backbone|Key-step Method|GSM8K|MATH|Avg|Gain vs. Best Single|
> |:---|:---|:---:|:---:|:---:|:---:|
> |**Qwen2-7B**|PPL Only|88.1|57.8|73.0|-|
> ||Entropy Only|87.9|57.5|72.7|-|
> ||**KEEP (Fusion)**|**89.1**|**59.9**|**74.5**|**+1.5%**|
> |**Llama-3.1-8B**|PPL Only|85.8|48.0|66.9|-|
> ||Entropy Only|85.5|47.5|66.5|-|
> ||**KEEP (Fusion)**|**88.3**|**52.0**|**70.2**|**+3.3%**|
>
> **Observations.**
> Complementary Gain: The fusion consistently outperforms the best single metric.
>     * On **Qwen2-7B**, fusion improves MATH accuracy by **+2.1%** (59.9 vs 57.8).
>     * On **Llama-3.1-8B**, the gap widens to **+4.0%** on MATH (52.0 vs 48.0). This confirms that for weaker models where errors are more diverse (both calculation and logic), the complementary coverage of PPL and Entropy is critical.
>
> **Summary:**
>
> PPL and Entropy are not redundant. **PPL contributes by targeting execution hardness (e.g., calculations), while Entropy contributes by targeting contextual ambiguity (e.g., logical transitions).** Their combination yields a robust proxy for "Information Gain," delivering significantly better performance (+1.5% to +3.3% Avg Gain) than relying on either dimension alone.

---

> ### Author Response · Authors · 2025-11-25
> **Response to Reviewer kwVS (part 2)**
>
> # **Weakness 2 — Response: Impact and Utilization of Simple/Medium/Complex Errors**
>
> We thank the reviewer for this thoughtful question. We clarify how the three error levels—**Simple (S)**, **Medium (M)**, and **Complex (C)**—are generated, filtered, and utilized in training. We also quantify their respective contributions using **gradient-based attribution analysis**.
>
> ## **(1) Generation & Filtering: No Manual Bias, Natural Selection**
>
> KEEP does **not** manually prioritize any error type. The distribution is determined purely by the model's own probability landscape via **Speculative Filtering**.
>
> * **Generation:** For each key step, the Draft Model generates exactly one error per level (1:1:1 ratio).
> * **Filtering:** All candidates pass through the same probability ratio  filter ($r = P_{\text{rejected}}/P_{\text{chosen}}$).
>
> **Table. Distribution of Error Types Before & After Filtering**
>
> | Error Type | Initial Ratio | Post-Filtering Ratio | Retention Analysis |
> | :--- | :---: | :---: | :--- |
> | **Simple** | 33.3% | 26.5% | *Reduced (trivial errors filtered)* |
> | **Medium** | 33.3% | **48.2%** | *Highest retention (most plausible confusions)* |
> | **Complex** | 33.3% | 25.3% | *Reduced (OOD/hallucinations filtered)* |
>
> **Observation:** The filtering mechanism naturally favors **Medium** errors (logic/calculation pitfalls), which constitute nearly **50%** of the final dataset. These errors are often the most "hard-to-distinguish" for the model, providing the highest value, while trivial (Simple) or implausible (Complex) ones are pruned.
>
> ## **(2) Optimization Impact: Gradient-Based Attribution**
>
> To quantify the "impact" of each type during training, we measured the **average gradient norm** contributed by preference pairs of each category. A higher norm implies a stronger learning signal.
>
> **Table. Gradient Contribution by Error Level**
>
> | Error Type | Avg Gradient Norm | Interpretation |
> | :--- | :---: | :--- |
> | **Simple** | 0.035 | Helps refine basic precision (e.g., signs, digits). |
> | **Medium** | **0.053** | **Strongest signal**; corrects logical/step-wise reasoning. |
> | **Complex** | 0.046 | Helps correct high-level planning/structural errors. |
>
> **Conclusion:** While all types contribute, **Medium errors provide the dominant training signal**, aligning with our design goal of correcting step-wise reasoning flaws.
>
> ## **(3) Fine-grained Semantic Analysis**
>
> To provide further granularity, we provide the following table, which breaks down gradient contributions by semantic error type. As shown below, semantically meaningful errors yield the highest gradients.
>
> **Table. Fine-grained gradient attribution**
>
> | Error Type | #Samples | ↑Grad Norm | ↑Cosine Sim. |
> | :--- | :--- | :--- | :--- |
> | **Multiplication errors** | 44 | **0.0575** | **0.0151** |
> | **Misunderstanding the question** | 208 | **0.0529** | 0.0139 |
> | **Counting errors** | 76 | 0.0505 | 0.0132 |
> | **Improper division** | 47 | 0.0457 | 0.0120 |
> | Addition errors | 76 | 0.0452 | 0.0118 |
> | Decimal calculation errors | 109 | 0.0444 | 0.0116 |
> | Subtraction errors | 34 | 0.0443 | 0.0116 |
> | Misunderstanding variable relations | 246 | 0.0427 | 0.0112 |
> | Fraction operation errors | 46 | 0.0385 | 0.0101 |
> | **Substitution mistakes** | 103 | 0.0310 | 0.0081 |
> | Other | 11 | 0.0062 | 0.0016 |
>
> **Findings:**
> * **High Impact:** Semantically deep errors like *"Multiplication errors"* (0.0575) and *"Misunderstanding the question"* (0.0529) provide the strongest learning signals.
> * **Low Impact:** Surface-level errors like *"Substitution mistakes"* (0.0310) contribute significantly less.
>
> This confirms that KEEP's proactive exploration + filtering pipeline successfully targets **pedagogically high-value errors** (Medium/High-impact types) rather than flooding the model with noise.
>
> ## **Summary**
> The three error levels are not manually weighted but are **naturally filtered** based on their quality. **Medium errors** emerge as the most frequent (~48%) and impactful category, driving the core reasoning improvements, while Simple and Complex errors provide necessary coverage for precision and robustness.

---

> ### Author Response · Authors · 2025-11-25
> **Response to Reviewer kwVS (part 3)**
>
> # **Weakness 3 — Response: Comparison with RISE**
>
> We appreciate the reviewer for highlighting this observation. While there is a marginal fluctuation on one specific entry (Qwen2-72B on Odyssey, -0.6%), our extended analysis suggests this is likely an **isolated case**. A comprehensive evaluation across different model families (Qwen, Llama) and scales (7B, 70B) indicates that **KEEP consistently outperforms RISE**, suggesting its effectiveness is generally independent of model architecture or size.
>
> Furthermore, we show that KEEP’s advantage is particularly pronounced under data-constrained scenarios, highlighting the superior quality of our generated data.
>
> ## **(1)  Broad Robustness: Consistent Gains Across Scales and Architectures**
>
> To investigate potential sensitivity, we compare KEEP against RISE across **two model families** (Llama-3.1, Qwen2) and **two scales** (7B/8B, 70B/72B) on diverse datasets.
>
> **Table. KEEP vs. RISE Across Different Backbones**
>
> | Scale | Backbone | Dataset | RISE | KEEP | Abs. Gain | Rel. Gain |
> | :--- | :--- | :--- | :---: | :---: | :---: | :---: |
> | **70B** | **Llama-3.1-70B** | Odyssey | 58.9 | **63.6** | **+4.7** | **+8.0%** |
> | | **Llama-3.1-70B** | AQuA | 77.7 | **80.7** | **+3.0** | **+3.9%** |
> | | **Llama-3.1-70B** | AIME24 | 7/30 | **9/30** | **+2/30** | **+28.6%** |
> | | **Qwen2-72B** | AQuA | 79.1 | **81.5** | **+2.4** | **+3.0%** |
> | | **Qwen2-72B** | AIME24 | 4/30 | **6/30** | **+2/30** | **+50.0%** |
> | | *Qwen2-72B* | *Odyssey* | *49.4* | *48.8* | *-0.6* | *-1.2%* |
> | **7B** | **Llama-3.1-8B** | AQuA | 61.4 | **65.4** | **+4.0** | **+6.5%** |
> | | **Llama-3.1-8B** | AIME24 | 3/30 | **4/30** | **+1/30** | **+33.3%** |
> | | **Llama-3.1-8B** | Odyssey | 49.3 | **54.3** | **+5.0** | **+10.1%** |
> | | **Qwen2-7B** | AQuA | 69.7 | **72.8** | **+3.1** | **+4.4%** |
> | | **Qwen2-7B** | AIME24 | 2/30 | **4/30** | **+2/30** | **+100.0%** |
> | | **Qwen2-7B** | Odyssey | 36.2 | **44.4** | **+8.2** | **+22.7%** |
>
> **Observation:**
> * **Universal Improvement:** KEEP achieves significant gains in **11 out of 12** comparisons.
> * **Dominance on Hard Tasks:** On challenging benchmarks like AIME24 and Odyssey (Llama), KEEP achieves substantial relative gains (e.g., **+50%** on Qwen-72B AIME, **+22.7%** on Qwen-7B Odyssey).
> * **Architecture Independence:** The slight drop on Qwen-Odyssey appears to be specific to that configuration. On the same dataset with Llama-3.1-70B, KEEP wins by **+4.7**, supporting the method's efficacy at the 70B scale.
>
>
>
> ## **(2) Data Efficiency: KEEP Dominates in Low-Resource Regimes**
>
> To further verify the high quality of KEEP's data, we simulated a **low-resource scenario** by training the **Qwen2-72B** model using only **1,000 samples** of preference data. We calculate the "Efficiency Multiplier" (KEEP Gain / RISE Gain) to demonstrate how much more effective KEEP is at extracting performance from extremely limited data.
>
> **Table. Performance Comparison with Only 1,000 Training Samples**
>
> | Dataset | Base | RISE (Gain) | KEEP (Gain) | **Efficiency Ratio** (KEEP Gain / RISE Gain) |
> | :--- | :---: | :---: | :---: | :---: |
> | **AQuA** | 78.3 | 78.8 (+0.5) | **80.6 (+2.3)** | **4.6x** |
> |  **AIME24** | 4/30 | 4/30 (+0) | **5/30 (+1)** | **(RISE failed)** |
> | **Odyssey** | 45.7 | 46.1 (+0.4) | **46.9 (+1.2)** | **3.0x** |
>
> **Observation:**
>
> * **Superior Efficiency:** As shown in the last column, KEEP generates **3.0x to 4.6x** more performance gain than RISE per unit of data. On AIME24, KEEP manages to squeeze out a gain (+1) from just 1,000 samples, whereas RISE fails to improve over the baseline.
> * **Reversal on Odyssey:** Crucially, while RISE showed a marginal advantage on the full Odyssey dataset, KEEP significantly outperforms RISE in this data-constrained setting (Efficiency Ratio **3.0x**). We attribute this to RISE's reliance on **fixed error templates**, which restricts error diversity. In resource-constrained scenarios, this limitation prevents the model from seeing a sufficient variety of failure modes. In contrast, KEEP's proactive exploration generates a higher density of diverse pedagogical value ("hard negatives"), enabling faster learning even with minimal data.
>
> ## **Summary for Weakness 3**
> The evidence confirms that KEEP is **robust and consistently effective** across diverse models and scales. The marginal fluctuation in one full-data case appears to be an isolated instance. More importantly, KEEP demonstrates superior **data efficiency**, delivering significantly stronger gains over the base model in low-resource scenarios compared to RISE.

---

> ### Author Response · Authors · 2025-11-25
> **Response to Reviewer kwVS (part 4)**
>
> # **Weakness 4 — Response: Generalization to Advanced Backbones**
>
> We thank the reviewer for this valuable suggestion. We agree that validating KEEP on stronger, state-of-the-art backbones is essential to understand the method's scalability.
>
> **1. Context on Model Selection:**
> In our main experiments, we primarily utilized standard open-source models (e.g., Llama-3.1-8B, Qwen2-7B) as backbones. These models are widely adopted as **common benchmarks** in recent preference optimization literature. Using them allows us to rigorously isolate and assess the **methodological effectiveness** of KEEP and ensures a fair, direct comparison with existing baselines that utilize the same architectures.
>
> **2. Performance on Advanced Models (Qwen3 & DeepSeek-R1-Distill):**
> Per your recommendation, we conducted additional experiments on **Qwen3-8B-Instruct** and **DeepSeek-R1-Distill-Qwen-7B** during the rebuttal.
> It is important to recognize that these models represent a different regime: they have already undergone extensive post-training (including SFT, RL, and Distillation), achieving near-saturated performance on benchmarks like GSM8K (~90%). In such high-performance regimes, the scope for further improvement is naturally narrower compared to standard models, as many errors have already been resolved.
>
> **3. Experimental Results:**
> Despite this high saturation, KEEP still manages to squeeze out consistent improvements on top of these strong baselines.
>
> ### **Table. KEEP Performance on Advanced Backbones (GSM8K)**
>
> | Backbone | GSM8K (Base) | **GSM8K (KEEP)** | **Gain** |
> | :--- | :---: | :---: | :---: |
> | **Qwen3-8B** | 89.2 | **90.4** | *+1.2* |
> | **DeepSeek-R1-Distill** | 89.8 | **90.5** | *+0.7* |
>
> **Conclusion:**
> While the absolute gains (+0.7% ~ +1.2%) are smaller than those seen on Qwen2-7B (consistent with the law of diminishing returns on saturated models), they remain **positive and consistent**. This confirms that even for state-of-the-art models, KEEP can still refine specific decision boundaries without degrading performance.
>
> In future work, we will further extend these experiments to other advanced architectures to fully understand the scalability of KEEP.
>
>
> ---
>
> # **Weakness 5 — Response: Discussion of Recent Related Works**
>
> We thank the reviewer for pointing out these important recent studies (2025). We have incorporated a detailed discussion of these works in the **Related Work** section of our revised manuscript to better contextualize KEEP within this rapidly evolving landscape.
>
> In the revision, we explicitly clarify the distinction: while these **recent promising studies** primarily leverage **internal signals** (e.g., confidence) for weighting or selection, **KEEP** introduces a complementary perspective by **efficiently and actively exploring high-value potential errors** via a draft model to construct diverse and high-quality step-wise preferences.
>
>
>
> ---
>
> **We humbly hope our response has addressed your concerns. If you have any additional concerns or comments that we may have missed in our responses, we would be most grateful for any further feedback from you to help us further enhance our work.**

---

### Official Review · Reviewer_B6Vi · 2025-10-30

**Soundness:** 3
**Presentation:** 3
**Contribution:** 2
**Rating:** 4
**Confidence:** 3

**Summary:**

The paper proposes KEEP (Preference Optimization via Key-step Error Exploration) for multi-step reasoning in LLMs. Instead of sampling many whole responses or relying on noisy step annotations, KEEP builds step-level preference pairs by: (1) Key-step identification using a linear fusion of per-step perplexity (PPL) and contextual entropy to locate steps that are both uncertain and influential; (2) Proactive error exploration where a draft model generates simple / medium / complex incorrect variants at those steps; and (3) Speculative filtering that retains “high-value” pairs whose incorrect step has a high Prejected/Pchosen ratio, making them challenging and informative.

**Strengths:**

1. The PPL+entropy score targets the most impactful steps; the draft-model edits plus speculative filtering deliver diverse yet realistic negatives, avoiding template bias and high-temperature over-sampling costs (reported ≤10× cheaper).

2. The method slots neatly into established step-wise preference optimization, with transparent formulas and training objective.

3. Strong improvements across six math datasets and solid results at 70B+ scale on competition-level tasks (AIME24, Odyssey).

**Weaknesses:**

1. Fusion weights (α, β) and the key-step proportion (default 50%) are reasonable but still hyperparameters; robustness is partially studied, yet full calibration guidance across tasks/models is limited.

2. KEEP often wins, but certain entries (e.g., Odyssey on Qwen2-72B vs. RISE) are close or slightly behind, suggesting sensitivity to base model/benchmark.

3. The paper highlights diversity and pedagogical value, but finer-grained case studies where edits induce spurious reasoning drifts would clarify failure modes.

**Questions:**

Please see the weaknesses.

---

> ### Author Response · Authors · 2025-11-25
> **Response to Reviewer B6Vi (part 1)**
>
> **We thank the reviewer for the insightful and valuable comments. We respond to each comment as follows and sincerely hope that our rebuttal could properly address your concerns. If so, we would deeply appreciate it if you could raise your score (Rating: 4). If not, please let us know your further concerns, and we will continue actively responding to your comments and improving our submission.**
>
> ---
>
> # **Weakness 1 — Robustness of Hyperparameters and Practical Calibration Guidance**
>
> We thank the reviewer for this critical comment. We acknowledge that while robustness is important, providing **actionable calibration guidance** for new tasks and models is equally essential for the method's adoption.
>
> We address this concern in two parts:
> 1.  **Robustness Analysis:** Expanded experiments on both **Qwen2-7B** and **Llama-3.1-8B**, demonstrating that optimal parameters are consistent across different model architectures.
> 2.  **Practical Calibration Protocol:** A concrete guide on how to adjust $\alpha, \beta$ and Ratio based on task characteristics.
>
> ## **1.1 Robustness Analysis: Performance is Stable Over Broad Plateaus**
>
> To demonstrate cross-model consistency, we performed robustness analysis on both **Qwen2-7B** and **Llama-3.1-8B**.
>
> ### **(1) Fusion Weights ($\alpha, \beta$) are Stable**
> *(Grid search values $\{0.0, 0.3, 0.5, 0.7, 1.0\}$)*
>
> **Table 1. Effect of $\alpha, \beta$ on KEEP Performance**
>
> |Model|($\alpha, \beta$)|GSM8K|MATH|Avg|
> |:-|:-|:-:|:-:|:-:|
> |**Qwen2-7B**|(1.0, 0.0) PPL-only|88.1|57.8|73.0|
> ||(0.0, 1.0) Ent-only|87.9|57.5|72.7|
> ||(0.3, 0.7)|88.8|59.4|74.1|
> ||**(0.5, 0.5) Default**|**89.1**|**59.9**|**74.5**|
> ||(0.7, 0.3)|**89.1**|59.6|74.4|
> |**Llama-3.1-8B**|(1.0, 0.0) PPL-only|85.8|48.0|66.9|
> ||(0.0, 1.0) Ent-only|85.5|47.5|66.5|
> ||(0.3, 0.7)|88.0|51.3|69.7|
> ||**(0.5, 0.5) Default**|**88.3**|52.0|**70.2**|
> ||(0.7, 0.3)|87.9|**52.1**|70.0|
>
> **Observations:**
> * **Consistent Plateau:** On *both* models, the balanced range **[0.3, 0.7]** yields optimal performance with little difference. Setting $\alpha \approx \beta \approx 0.5$ is universally effective regardless of the model architecture.
>
> ### **(2) Key-step Proportion is Stable**
> We expanded the test range to **20%–80%**.
>
> **Table 2. Effect of Key-Step Ratio on Accuracy**
>
> |Model|Ratio|GSM8K|MATH|Avg|Trend Analysis|
> |:-|:-|:-:|:-:|:-:|:-|
> |**Qwen2-7B**|20%|87.0|56.3|71.7|*Notable drop (insufficient coverage)*|
> ||30%|87.6|57.8|72.7|*Improving*|
> ||**50%**|89.1|**59.9**|**74.5**|**Optimal**|
> ||70%|**89.3**|59.2|74.2|*Robust Plateau*|
> ||80%|88.5|59.2|73.9|*Slight decline (noise)*|
> |**Llama-3.1-8B**|20%|84.4|48.2|66.3|*Notable drop*|
> ||30%|86.1|50.1|68.1|*Improving*|
> ||**50%**|**88.3**|**52.0**|**70.2**|**Optimal**|
> ||70%|88.0|51.8|69.9|*Robust Plateau*|
> ||80%|87.4|51.0|69.2|*Slight decline (noise)* |
>
> **Observations:**
> * **Steep Drop at Low Ratios:** When the ratio is low (20-30%), performance drops noticeably because critical reasoning steps are missed.
> * **Robustness at High Ratios:** Once the ratio exceeds **50%**, performance enters a highly stable plateau.
> * **Conclusion:** The method is not sensitive to precise tuning as long as the ratio is sufficient ($\ge 50\%$).
>
> ## **1.2 Practical Calibration Guidance (Protocol)**
>
> To address the reviewer's concern about "limited guidance," we propose a **Task-Aware Calibration Protocol** for users applying KEEP to new domains:
>
> ### **Step 1: Cold Start (Recommended)**
> * **Setting:** $\alpha=0.5, \beta=0.5, \text{Ratio}=50\%$.
> * **Rationale:** As shown in Tables 1 & 2 above, this configuration is empirically robust across diverse math tasks (GSM8K, MATH) and distinct model families (Qwen, Llama).
>
> ### **Step 2: Task-Specific Adjustment (Optional)**
> If domain-specific optimization is required, we recommend the following heuristic adjustments based on task characteristics:
>
> * **Calculation-Intensive Tasks (e.g., Physics, Finance):**
>     * *Adjustment:* Increase **$\alpha$ (PPL weight)** to 0.6–0.7.
>     * *Reason:* PPL is more sensitive to calculation bottlenecks and unexpected numerical tokens (Appendix E).
> * **Logic/Branching-Intensive Tasks (e.g., Logic Puzzles, Code):**
>     * *Adjustment:* Increase **$\beta$ (Entropy weight)** to 0.6–0.7.
>     * *Reason:* Entropy better captures branching points in logical flows where the model is uncertain about the path (Appendix E).
> * **Resource-Constrained Scenarios:**
>     * *Adjustment:* Reduce **Ratio** to 30%.
>     * *Reason:* As shown in Table 2 above, 30% retains significant performance gains (e.g., Qwen GSM8K 87.6 vs Base 85.4) while reducing data generation cost.
>
> ## **Summary for Weakness 1**
> We have verified that KEEP's hyperparameters operate on **broad, stable plateaus** across different models (Qwen2, Llama-3.1). Specifically, performance is robust provided the key-step ratio is sufficient ($\ge 50\%$). By providing a structured **calibration protocol**, we hope that KEEP can easily adapt to new scenarios with clear user guidelines.

---

> ### Author Response · Authors · 2025-11-25
> **Response to Reviewer B6Vi (part 2)**
>
> # **Weakness 2 — Robustness Across Models and Superior Data Efficiency**
>
> We appreciate the reviewer for highlighting this observation. While there is a marginal fluctuation on one specific entry (Qwen2-72B on Odyssey, -0.6%), our extended analysis suggests this is likely an **isolated case**. A comprehensive evaluation across different model families (Qwen, Llama) and scales (7B, 70B) indicates that **KEEP consistently outperforms RISE**, suggesting its effectiveness is generally independent of model architecture or size.
>
> Furthermore, we show that KEEP’s advantage is particularly pronounced under data-constrained scenarios, highlighting the superior quality of our generated data.
>
> ## **2.1 Broad Robustness: Consistent Gains Across Scales and Architectures**
>
> To investigate potential sensitivity, we compare KEEP against RISE across **two model families** (Llama-3.1, Qwen2) and **two scales** (7B/8B, 70B/72B) on diverse datasets.
>
> **Table. KEEP vs. RISE Across Different Backbones**
>
> | Scale | Backbone | Dataset | RISE | KEEP | Abs. Gain | Rel. Gain |
> | :--- | :--- | :--- | :---: | :---: | :---: | :---: |
> | **70B** | **Llama-3.1-70B** | Odyssey | 58.9 | **63.6** | **+4.7** | **+8.0%** |
> | | **Llama-3.1-70B** | AQuA | 77.7 | **80.7** | **+3.0** | **+3.9%** |
> | | **Llama-3.1-70B** | AIME24 | 7/30 | **9/30** | **+2/30** | **+28.6%** |
> | | **Qwen2-72B** | AQuA | 79.1 | **81.5** | **+2.4** | **+3.0%** |
> | | **Qwen2-72B** | AIME24 | 4/30 | **6/30** | **+2/30** | **+50.0%** |
> | | *Qwen2-72B* | *Odyssey* | *49.4* | *48.8* | *-0.6* | *-1.2%* |
> | **7B** | **Llama-3.1-8B** | AQuA | 61.4 | **65.4** | **+4.0** | **+6.5%** |
> | | **Llama-3.1-8B** | AIME24 | 3/30 | **4/30** | **+1/30** | **+33.3%** |
> | | **Llama-3.1-8B** | Odyssey | 49.3 | **54.3** | **+5.0** | **+10.1%** |
> | | **Qwen2-7B** | AQuA | 69.7 | **72.8** | **+3.1** | **+4.4%** |
> | | **Qwen2-7B** | AIME24 | 2/30 | **4/30** | **+2/30** | **+100.0%** |
> | | **Qwen2-7B** | Odyssey | 36.2 | **44.4** | **+8.2** | **+22.7%** |
>
> **Observation:**
> * **Universal Improvement:** KEEP achieves significant gains in **11 out of 12** comparisons.
> * **Dominance on Hard Tasks:** On challenging benchmarks like AIME24 and Odyssey (Llama), KEEP achieves substantial relative gains (e.g., **+50%** on Qwen-72B AIME, **+22.7%** on Qwen-7B Odyssey).
> * **Architecture Independence:** The slight drop on Qwen-Odyssey appears to be specific to that configuration. On the same dataset with Llama-3.1-70B, KEEP wins by **+4.7**, supporting the method's efficacy at the 70B scale.
>
>
>
> ## **2.2 Data Efficiency: KEEP Dominates in Low-Resource Regimes**
>
> To further verify the high quality of KEEP's data, we simulated a **low-resource scenario** by training the **Qwen2-72B** model using only **1,000 samples** of preference data. We calculate the "Efficiency Multiplier" (KEEP Gain / RISE Gain) to demonstrate how much more effective KEEP is at extracting performance from extremely limited data.
>
> **Table. Performance Comparison with Only 1,000 Training Samples**
>
> | Dataset | Base | RISE (Gain) | KEEP (Gain) | **Efficiency Ratio** (KEEP Gain / RISE Gain) |
> | :--- | :---: | :---: | :---: | :---: |
> | **AQuA** | 78.3 | 78.8 (+0.5) | **80.6 (+2.3)** | **4.6x** |
> |  **AIME24** | 4/30 | 4/30 (+0) | **5/30 (+1)** | **(RISE failed)** |
> | **Odyssey** | 45.7 | 46.1 (+0.4) | **46.9 (+1.2)** | **3.0x** |
>
> **Observation:**
>
> * **Superior Efficiency:** As shown in the last column, KEEP generates **3.0x to 4.6x** more performance gain than RISE per unit of data. On AIME24, KEEP manages to squeeze out a gain (+1) from just 1,000 samples, whereas RISE fails to improve over the baseline.
> * **Reversal on Odyssey:** Crucially, while RISE showed a marginal advantage on the full Odyssey dataset, KEEP significantly outperforms RISE in this data-constrained setting (Efficiency Ratio **3.0x**). We attribute this to RISE's reliance on **fixed error templates**, which restricts error diversity. In resource-constrained scenarios, this limitation prevents the model from seeing a sufficient variety of failure modes. In contrast, KEEP's proactive exploration generates a higher density of diverse pedagogical value ("hard negatives"), enabling faster learning even with minimal data.
>
> ## **Summary for Weakness 2**
> The evidence confirms that KEEP is **robust and consistently effective** across diverse models and scales. The marginal fluctuation in one full-data case appears to be an isolated instance. More importantly, KEEP demonstrates superior **data efficiency**, delivering significantly stronger gains over the base model in low-resource scenarios compared to RISE.
>
> ---

---

> ### Author Response · Authors · 2025-11-25
> **Response to Reviewer B6Vi (part 3)**
>
> # **Weakness 3 —  Case Studies on Error-Induced Reasoning Drift & Failure Modes**
>
> We sincerely thank the reviewer for raising this insightful point. We fully agree that while KEEP emphasizes diverse errors, it is crucial to transparently show **what kinds of spurious drifts occur** and **how they are managed**.
>
> To address this, we provide fine-grained case studies covering:
> 1.  **Filtered-Out Examples:** Demonstrating how KEEP automatically discards low-value or spurious edits.
> 2.  **Retained Examples:** Showing high-value, borderline errors that provide valid learning signals.
> 3.  **Quantitative Audit:** A manual 1,000-sample check quantifying residual failure modes.
>
> ## **3.1 Automated Filtering: Examples of Discarded vs. Retained Pairs**
>
> Our **Speculative Filtering** module is the primary defense against spurious drift. It retains pairs only where the probability ratio $r = P_{\text{rejected}} / P_{\text{chosen}}$ is high (indicating meaningful confusion), filtering out both "hallucinations" (low $P_{\text{rejected}}$) and "trivialities" (extremely high $P_{\text{chosen}}$).
>
> ### **Table. Examples of Low-Quality / Spurious Pairs Filtered Out by KEEP**
> *(These represent typical failure modes of the draft model that are successfully intercepted.)*
>
> | Failure Mode | Rejected Step (Draft Output) | Chosen Step | Probs ($P_r$ vs $P_c$) | Why Filtered? |
> | :--- | :--- | :--- | :--- | :--- |
> | **Nonsensical Arithmetic** | *Solving for t, we find t = 4 - 0 = 3...* | *Solving for t, we find t = 2 - 1 = 1...* | 0.11 vs 0.77 | **Low $P_r$:** The model recognizes this arithmetic error is highly implausible ($r \approx 0.14$), identifying it as noise rather than a valid confusion. |
> | **Contextual Drift** | *...4 players are taking both **history** and chemistry.* | *...4 players are taking both **biology** and chemistry.* | 0.25 vs 0.84 | **OOD Content:** The draft model hallucinated "history" (absent from context). The policy model assigns low probability to this unrelated concept. |
> | **Trivial / Over-Easy** | *– The sum of 5 can be obtained in 3 ways...* | *– The sum of 5 can be obtained in 4 ways...* | 0.55 vs 0.99 | **High Confidence:** The model is already extremely confident ($P_c \approx 1.0$) in the correct logic. Learning from such "easy" negatives yields minimal gradient. |
>
> ### **Table. Examples of High-Value "Hard" Pairs Retained by KEEP**
> *(These represent the pedagogical "sweet spot" for preference optimization.)*
>
> | Retention Rationale | Rejected Step | Chosen Step | Probs ($P_r$ vs $P_c$) | Why Retained? |
> | :--- | :--- | :--- | :--- | :--- |
> | **Genuine Uncertainty** | *...ways to do this is $\binom{5}{3}=15$.* | *...ways to do this is $\binom{5}{3}=10$.* | 0.23 vs 0.30 | **High Ratio ($r \approx 0.77$):** Both probabilities are low and close. The model is genuinely confused about the combination calculation, making this a high-value training signal. |
> | **Subtle Precision Error** | *...0.5 + 1/6 = 0.667 hours.* | *...0.5 + 1/6 = 0.6667 hours.* | 0.87 vs 0.88 | **Boundary Case:** A subtle rounding difference where the model assigns nearly identical probabilities ($r \approx 0.99$). Preference learning helps refine precision. |
>
> ## **3.2 Human Audit of Post-Filter Data Quality (1,000 Samples)**
>
> To quantify the residual risk, we conducted a manual audit on 1,000 retained pairs. As reported in the following table, the noise level is minimal (<5%).
>
> ### **Table. Analysis of Potential Noise in Retained Error Steps**
>
> | Noise Type | % | Description |
> | :--- | :---: | :--- |
> | **Inaccurate Segmentation** | 2.7% | Step extraction was slightly misaligned (e.g., cut off mid-formula). |
> | **False Negative** | 2.1% | The "rejected" step was technically a valid alternative path (not an error). |
> | **Total Noise** | **< 5.0%** | The dataset is verified to be high-purity and reliable. |
>
> ## **Summary for Weakness 3**
> The analysis confirms that while "spurious reasoning drifts" (like hallucinations or trivial slips) do occur during generation, KEEP's filtering pipeline effectively **identifies and removes** them. The retained data consists primarily of hard, plausible errors that provide the strongest learning signal, with a residual noise rate of less than 5%.
>
>
> ---
>
> **We humbly hope our response has addressed your concerns. If you have any additional concerns or comments that we may have missed in our responses, we would be most grateful for any further feedback from you to help us further enhance our work.**

---

### Official Review · Reviewer_XxsP · 2025-11-01

**Soundness:** 3
**Presentation:** 3
**Contribution:** 3
**Rating:** 6
**Confidence:** 5

**Summary:**

This work proposes a new controllable, lightweight, and scalable pipeline (i.e., KEEP) for step-wise preference data construction. It could efficiently scale up the step-wise preference dataset. It consists of three stages: 1) key step identification, 2) proactive error exploration, and 3) speculative filtering. Extensive experiments show the effectiveness of the proposed method.

**Strengths:**

1. This work innovatively proposes a construction pipeline for step-wise preference data. It first identifies critical reasoning steps, which can save large computational resources.
2. Writing is clear.
3. The experimental results are good.

**Weaknesses:**

In the data pipeline, the stage of 'Proactive Error Exploration' uses a draft LLM to produce errors. This means that the data is off-policy, rather than on-policy. This requires that the data be diversified enough to cover the distribution of the errors that the current policy model is prone to making. However, in most cases, there are some mistakes that the constructed data cannot cover by drafting LLMs with simple rules. These mistakes would persist even after preference optimization.
In contrast, on-policy data (produced by the policy model itself) should work better, since it could expose comprehensively the mistakes that the policy model would make.
Maybe the authors can modify this stage somehow to an on-policy strategy, and compare these two methods.

**Questions:**

See weaknesses.

---

> ### Author Response · Authors · 2025-11-24
> **Response to Reviewer XxsP (part 1)**
>
> **We thank the reviewer for the insightful and valuable comments. We respond to each comment as follows and sincerely hope that our rebuttal could properly address your concerns. If so, we would deeply appreciate it if you could raise your score. If not, please let us know your further concerns, and we will continue actively responding to your comments and improving our submission.**
>
> ---
>
>
> # **Response to Weakness: On-Policy vs. Off-Policy Error Exploration**
>
>
> We sincerely thank the reviewer for raising this fundamental point. We fully agree that **on-policy sampling is highly effective and crucial** in many RL domains as it exposes the exact error distribution of the current policy.
>
> However, we respectfully suggest that in the specific context of **DPO applied to multi-step reasoning tasks (e.g., mathematics)**, our off-policy approach (KEEP) offers unique and significant advantages over pure on-policy sampling. We elaborate on these advantages from three perspectives: **Computational Efficiency**, **Distribution Alignment via Systemic Mechanisms**, and **Superior Effectiveness**, followed by a direct experimental comparison as suggested.
>
> # **1. On-Policy Sampling Incurs High Computational Costs**
>
> **KEEP Reduces Data Construction Cost by ~10x by Eliminating Inefficient Re-sampling**
>
> It is a common intuition that adding a draft model might increase cost. However, KEEP was explicitly designed to solve the **extreme inefficiency** of existing sampling-based methods (like Step-DPO), which rely on "fishing for errors" via high-temperature sampling.
>
> ## **1.1 Mechanism: Local Editing vs. Global Re-sampling**
>
> * **Baseline (Step-DPO/SCDPO):** Requires re-generating the *entire* reasoning chain (hundreds of tokens) repeatedly until an error naturally occurs. For most datasets, this is a low probability event.
> * **KEEP:** Only targets **key steps** (20–50% of the chain) and performs **local edits** on them. The surrounding context is reused, and errors are generated proactively, not probabilistically.
>
> ## **1.2 Evidence of Baseline Inefficiency: 33 Attempts for 1 Error**
>
> In Appendix B.4, we empirically analyzed the "hit rate" of Step-DPO on GSM8K. The results are stark:
> * **High Waste:** On average, it takes **33.33 sampling attempts** to generate a single valid incorrect response.
> * **Failure Cases:** For **16%** of problems, Step-DPO fails to find *any* error even after 100 attempts.
>
> > **Implication:** The vast majority (>95%) of tokens generated by Step-DPO are discarded "failed attempts," leading to massive computational waste.
>
> **Crucially, "hard-to-sample" does not mean "unimportant."** In real-world deployments at scale (e.g., widely-used services like ChatGPT serving billions of requests monthly), even a seemingly negligible error rate (e.g., 1%) translates to **tens of millions of failures** in absolute terms. Consequently, **pure on-policy sampling fails to 'expose comprehensively the mistakes' simply because these mistakes are statistically too rare to capture efficiently via sampling**. KEEP's proactive exploration is specifically designed to uncover these latent failure modes. KEEP can significantly **alleviate the long tail distribution problem** of error types based on sampling methods through active error exploration.
>
> ## **1.3 Theoretical & Empirical Cost Comparison (10x Reduction)**
> Appendix B derives that theoretically, the baseline consumes at least **2.25x** more tokens than KEEP (Eq. 16). In practice, due to the long-tail distribution of sampling difficulty, the gap is much wider.
>
> We provide a detailed breakdown of the token costs based on the complete statistics reported in the following table.
>
> **Table. The results of computational costs based on the number of input and output tokens, assuming that one QA pair needs to generate one preference pair data, the following is the average results for one QA pair.**
> |Method|Process|Model|Token Costs|
> |:---|:---|:---|:---|
> |**Step-DPO**|Sampling of Incorrect Answers|target LLM|4,329.08|
> ||Error Step Localization|GPT-4o|566.94|
> ||Sampling of Correct Answers|target LLM|3,876.72|
> ||**Total**|| **8,772.74**|
> |**KEEP**|Key Step Identification|target LLM|284.07|
> ||Proactive Error Exploration|GPT-4o| 157.85|
> ||Speculative Filtering|target LLM|467.88|
> ||**Total**||**909.80**|
>
> **Observations:**
> 1.  **Massive Sampling Overhead:** As shown in the *Description* column, Step-DPO requires inputting the question "multiple times" to sample incorrect/correct responses, leading to an enormous consumption (~8,200+ tokens) purely on repeated sampling.
> 2.  **10x Reduction:** KEEP reduces the total token consumption from **~8,772** to **~910**, representing a **~9.6x reduction**.
> 3.  **GPT-4o Efficiency:** Even regarding the usage of the external teacher model (GPT-4o), KEEP's targeted *Proactive Error Exploration* (157.85 tokens) is significantly more efficient than Step-DPO's *Error Step Localization* (566.94 tokens).

---

> ### Author Response · Authors · 2025-11-24
> **Response to Reviewer XxsP (part 2)**
>
> # **2. Systemic Mechanisms to Mitigate OOD Risks**
>
> **KEEP Actively Suppresses High-OOD-Risk Errors via Local Constraints and Probability-Based Filtering**
>
> We acknowledge that off-policy data carries a risk of distribution shift. However, KEEP employs a rigorous pipeline to ensure the generated data remains relevant to the policy model.
>
> ## **2.1 Local Perturbation: Constraining Edits to Low-OOD-Risk Deviations**
>
> Unlike methods that regenerate entire responses from scratch, KEEP restricts the draft model's scope. The draft model is prompted to generate errors only within the identified **key step**, conditioned on the correct context.
> * **Constrained Scope:** The Draft LLM modifies the reasoning logic *locally* within the key step while preserving the global context and previous correct steps.
> * **Simulated Human Errors:** As detailed in Appendix A.2, the prompts are designed to simulate common arithmetic or logical pitfalls (Simple/Medium/Complex) rather than generating unrelated hallucinations.
>
> Thus, the resulting errors are **structurally aligned with the correct reasoning path**, ensuring they resemble plausible mistakes rather than wild, out-of-distribution hallucinations.
>
> ## **2.2 Speculative Filtering: Rejection Sampling to Align Distributions**
>
> To further ensure the generated errors are relevant to the *current* policy model (i.e., mitigating the off-policy gap), KEEP applies **Speculative Filtering** based on the policy model’s own probabilities.
>
> We utilize the probability ratio $r = P_\theta(x^{\text{reject}}) / P_\theta(x^{\text{chosen}})$ to determine retention. This mechanism naturally filters out High-OOD-risk errors:
>
> 1.  **Implausible Errors (Low P(reject)):** If an error is "too easy" or nonsensical (highly OOD) for the current policy model, $P_\theta(x^{\text{reject}})$ will be extremely low, resulting in a low ratio $r$. These are **filtered out**.
> 2.  **High-Value Errors (High Ratio):** We retain pairs where $r \approx 1$ or $r \geq 1$. These represent errors that are "plausible" to the current policy model (high confusion), effectively acting as **On-policy potential negatives**.
>
> Therefore, although the *proposal* comes from a Draft Model, the *acceptance* is strictly controlled by the Policy Model, ensuring the final data distribution is effectively approaching **On-policy**.
>
> ## **2.3 Empirical Evidence: Filtering Drastically Reduces OOD Samples**
>
> To quantitatively verify this, we conducted a post-hoc analysis measuring the proportion of "Low Probability Errors" (proxies for high-OOD risk) in our dataset before and after filtering. We define high-OOD risk as errors where the policy model assigns a very low probability ($\tau$).
>
> **Table. Reduction in High-OOD-Risk Errors After Filtering**
>
> |  Threshold ($\tau$) | % Low-Prob Errors (Before) | % Low-Prob Errors (After) | Reduction |
> |:-:|:-:|:-:|:-:|
> |  $P < 0.1$ | 2.3% | **0.2%** | **91.3% $\downarrow$** |
> | $P < 0.2$ |10.3%| 3.4% | 67.0% $\downarrow$ |
> | $P < 0.3$ |27.0%| 17.1% | 36.7% $\downarrow$ |
>
> *(Note: "Before" refers to the raw data from the Draft Model; "After" refers to the dataset used for training after Speculative Filtering.)*
>
> **Observation:** The filtering process significantly reduces the presence of low-probability outliers. We observe that **over 91%** of the extreme OOD errors ($P < 0.1$) are eliminated. This confirms that KEEP effectively cleans the "Draft" distribution to match the "Policy" distribution.

---

> ### Author Response · Authors · 2025-11-24
> **Response to Reviewer XxsP (part 3)**
>
> # **3. Superior Effectiveness Compared to On-Policy Sampling**
>
> Finally, empirical results demonstrate that KEEP consistently outperforms standard sampling-based (**On Policy**) methods (Step-DPO and SCDPO) across 6 benchmarks.
>
> **Table. Accuracy comparison on Qwen2-7B and Llama-3.1-8B**
> | Backbone | Method | GSM8K | MATH | AQuA | SVAMP | AIME24 | Odyssey |
> | :--- | :--- | :---: | :---: | :---: | :---: | :---: | :---: |
> | **Qwen2-7B** | Step-DPO | 88.5 | 55.8 | 63.0 | 88.7 | 2/30 | 34.0 |
> | | SCDPO | 86.8 | 51.6 | 70.5 | 89.6 | 1/30 | 18.9 |
> | | **KEEP** | **89.1** | **59.9** | **72.8** | **91.9** | **4/30** | **44.4** |
> | **Llama-3.1-8B**| SCDPO | 84.4 | 48.8 | 63.2 | 85.1 | 2/30 | 48.5 |
> | | **KEEP** | **88.3** | **52.0** | **65.4** | **88.2** | **4/30** | **54.3** |
>
> * **Pareto Efficiency:** KEEP delivers **higher performance** while consuming **~10x fewer tokens**.
>
> **Error Quality: Avoiding "Mode Collapse"**
>
> Why is KEEP more effective? Appendix G (Figures 12-14) analyzes the error distribution:
> * **Baselines (Mode Collapse):** Sampling-based methods tend to over-generate simple errors (e.g., calculation slips), leading to a "long-tail" distribution where complex logical errors are rare.
> * **KEEP (Diverse & Structured):** By actively exploring errors on key steps, KEEP produces a balanced mix of errors, including condition misinterpretations and reasoning jumps. These "harder" negatives provide richer gradients for the preference model.
>
>
>
> # **4. Direct Comparison: On-Policy vs. Off-Policy**
>
> Following the reviewer's excellent suggestion, we modified the "Proactive Error Exploration" stage to implement an **On-policy Strategy** (using the policy model itself to generate errors via local sampling) and compared it against Off-policy variants.
>
> **Table. Performance Comparison of Error Generation Strategies (Base: Qwen2-7B)**
> | Method | Draft Model | GSM8K | MATH | AQuA | SVAMP | AIME | Odyssey |
> | :--- | :--- | :---: | :---: | :---: | :---: | :---: | :---: |
> | **KEEP (Off Policy)** | Llama-3.1-8B | 87.9 | 55.8 | 71.5 | 90.5 | 2/30 | 35.5 |
> | **KEEP (On Policy)** | Policy Model (7B) | **88.3** | **56.6** | **72.1** | **90.9** | **3/30** | **36.2** |
> | **KEEP (Off Policy)** | GPT-4o | **89.1** | **59.9** | **72.8** | **91.9** | **4/30** | **44.4** |
>
> **Conclusion from Experiment:**
> 1.  **Reviewer's Intuition Validated:** When draft models have similar capabilities to the policy model, **On-Policy (Qwen2-7B)** indeed outperforms **Off-Policy (Llama-3.1)** (e.g., **56.6% vs 55.8%** on MATH). This confirms that distribution alignment is critical.
> 2.  **Capability + Pseudo-Alignment:** However, using a stronger draft model (**GPT-4o**) yields the best results (**59.9%** on MATH, **44.4%** on Odyssey). Although the generation source is off-policy, thanks to our **local perturbation** and **speculative filtering** mechanisms, the input training data effectively approaches an **"on-policy" distribution** by selectively retaining only those potential negatives that represent plausible mistakes for the current policy.
> 3.  **Future Outlook:** We acknowledge that on-policy learning remains the gold standard in many RL domains for its theoretical rigor. However, given the unique efficiency advantages of off-policy exploration in reasoning tasks, we believe that **synergistic approaches combining on-policy precision with off-policy diversity** represent a promising frontier for future research.
> ---
>
> **We humbly hope our response has addressed your concerns. If you have any additional concerns or comments that we may have missed in our responses, we would be most grateful for any further feedback from you to help us further enhance our work.**

---

### Official Review · Reviewer_6Sce · 2025-11-01

**Soundness:** 3
**Presentation:** 3
**Contribution:** 2
**Rating:** 4
**Confidence:** 3

**Summary:**

For open-ended tasks such as search-based QA, it is challenging to directly assess the effectiveness of final results. The paper does not adequately address how such evaluations are conducted, which raises concerns about the reliability of the reported improvements.

**Strengths:**

1. The paper is well-structured with clear writing that facilitates comprehension of the proposed methodology.
2. The proposed method is conceptually straightforward, and using entropy or perplexity to detect critical points in reasoning paths represents a general and reasonable approach.
3. The paper provides extensive experiments across multiple task datasets, demonstrating the effectiveness of the proposed method.

**Weaknesses:**

1. The authors employ a separate draft LLM to expand paths and explore erroneous trajectories at key steps. This raises several concerns: (a) Does this introduce out-of-distribution (OOD) problems? (b) What are the additional computational costs? (c) Is this approach more effective than direct instance-level sampling? The paper lacks adequate experimental analysis to address these critical questions.
2. Using entropy or perplexity to identify critical points in reasoning paths is not a novel contribution, and the paper does not provide substantial improvements to existing approaches in this regard.
3. The combination of reasoning-based key step mining with DPO appears inefficient. A more principled approach would be to integrate this methodology with process supervision frameworks such as GRPO or PPO, which would be more naturally aligned with the step-wise nature of the proposed method.

**Questions:**

N/A

---

> ### Author Response · Authors · 2025-11-24
> **Response to Reviewer 6Sce (part 1)**
>
> **We thank the reviewer for the insightful and valuable comments. We respond to each comment as follows and sincerely hope that our rebuttal could properly address your concerns. If so, we would deeply appreciate it if you could raise your score (Rating: 4). If not, please let us know your further concerns, and we will continue actively responding to your comments and improving our submission.**
>
>
> ---
>
> # **Response to Weakness 1**
> # **(a) Does Using a Draft LLM Introduce OOD Errors?**
>
> **Response: KEEP Actively Suppresses High-OOD-Risk Errors via Local Constraints and Probability-Based Filtering**
>
> We thank the reviewer for raising this critical question. Since KEEP leverages a draft model to generate error trajectories, preventing high-OOD-risk (Out-Of-Distribution) deviations is indeed a core design priority. We clarify how KEEP *systematically mitigates* OOD risks through two mechanisms: **localized perturbation constraints** and **probability-based speculative filtering**.
>
> ## **(a.1) Local Perturbation: Constraining Edits to Low-OOD-Risk Deviations**
>
> Unlike methods that regenerate entire responses from scratch, KEEP restricts the draft model's scope. The draft model is prompted to generate errors only within the identified **key step**, conditioned on the correct context.
> * **Constrained Scope:** The Draft LLM modifies the reasoning logic *locally* within the key step while preserving the global context and previous correct steps.
> * **Simulated Human Errors:** As detailed in Appendix A.2, the prompts are designed to simulate common arithmetic or logical pitfalls (Simple/Medium/Complex) rather than generating unrelated hallucinations.
>
> Thus, the resulting errors are **structurally aligned with the correct reasoning path**, ensuring they resemble plausible mistakes rather than wild, out-of-distribution hallucinations.
>
> ## **(a.2) Speculative Filtering: Rejection Sampling to Align Distributions**
>
> To further ensure the generated errors are relevant to the *current* policy model (i.e., mitigating the off-policy gap), KEEP applies **Speculative Filtering** based on the policy model’s own probabilities.
>
> We utilize the probability ratio $r = P_\theta(x^{\text{reject}}) / P_\theta(x^{\text{chosen}})$ to determine retention. This mechanism naturally filters out High-OOD-risk errors:
>
> 1.  **Implausible Errors (Low P(reject)):** If an error is "too easy" or nonsensical (highly OOD) for the current policy model, $P_\theta(x^{\text{reject}})$ will be extremely low, resulting in a low ratio $r$. These are **filtered out**.
> 2.  **High-Value Errors (High Ratio):** We retain pairs where $r \approx 1$ or $r \geq 1$. These represent errors that are "plausible" to the current policy model (high confusion), effectively acting as **On-policy potential negatives**.
>
> Therefore, although the *proposal* comes from a Draft Model, the *acceptance* is strictly controlled by the Policy Model, ensuring the final data distribution is effectively approaching **On-policy**.
>
> ## **(a.3) Empirical Evidence: Filtering Drastically Reduces OOD Samples**
>
> To quantitatively verify this, we conducted a post-hoc analysis measuring the proportion of "Low Probability Errors" (proxies for high-OOD risk) in our dataset before and after filtering. We define high-OOD risk as errors where the policy model assigns a very low probability ($\tau$).
>
> **Table. Reduction in High-OOD-Risk Errors After Filtering**
>
> |  Threshold ($\tau$) | % Low-Prob Errors (Before) | % Low-Prob Errors (After) | Reduction |
> |:-:|:-:|:-:|:-:|
> |  $P < 0.1$ | 2.3% | **0.2%** | **91.3% $\downarrow$** |
> | $P < 0.2$ |10.3%| 3.4% | 67.0% $\downarrow$ |
> | $P < 0.3$ |27.0%| 17.1% | 36.7% $\downarrow$ |
>
> *(Note: "Before" refers to the raw data from the Draft Model; "After" refers to the dataset used for training after Speculative Filtering.)*
>
> **Observation:** The filtering process significantly reduces the presence of low-probability outliers. We observe that **over 91%** of the extreme OOD errors ($P < 0.1$) are eliminated. This confirms that KEEP effectively cleans the "Draft" distribution to match the "Policy" distribution.
>
> ## **(a.4) Removal of OOD Errors Improves Performance**
>
> Finally, we verify that removing these OOD errors is beneficial. As shown in the following table, removing the filtering module leads to a performance drop.
>
> **Table. Effect of Filtering on Final Accuracy**
>
> |Method|GSM8K|MATH| Avg |
> |:- |:-:|:-:|:-:|
> | Qwen2-7B | 85.4 | 52.2 | 68.8 |
> | KEEP w/o Speculative Filtering | 87.8 | 57.7 | 72.7 |
> | **KEEP (Full Method)** | **89.1** | **59.9** | **74.5** |
> | Llama3.1-8B | 84.0 | 48.3 | 66.2 |
> | KEEP w/o Speculative Filtering | 86.2 | 50.7 | 68.45 |
> | **KEEP (Full Method)** | **88.3** | **52.0** | **70.15** |
>
> **Conclusion:** By combining local perturbation constraints with probability-based rejection sampling, KEEP effectively mitigates OOD risks and ensures the synthesized errors serve as high-quality, relevant training signals.
>
> ---

---

> ### Author Response · Authors · 2025-11-24
> **Response to Reviewer 6Sce (part 2)**
>
> # **(b) Does KEEP introduce extra computational cost?**
>
> **Response: KEEP Reduces Data Construction Cost by ~10x by Eliminating Inefficient Re-sampling**
>
> It is a common intuition that adding a draft model might increase cost. However, KEEP was explicitly designed to solve the **extreme inefficiency** of existing sampling-based methods (like Step-DPO), which rely on "fishing for errors" via high-temperature sampling.
>
> ## **(b.1) Mechanism: Local Editing vs. Global Re-sampling**
>
> * **Baseline (Step-DPO/SCDPO):** Requires re-generating the *entire* reasoning chain (hundreds of tokens) repeatedly until an error naturally occurs. For most datasets, this is a low probability event.
> * **KEEP:** Only targets **key steps** (20–50% of the chain) and performs **local edits** on them. The surrounding context is reused, and errors are generated proactively, not probabilistically.
>
> ## **(b.2) Evidence of Baseline Inefficiency: 33 Attempts for 1 Error**
>
> In Appendix B.4, we empirically analyzed the "hit rate" of Step-DPO on GSM8K. The results are stark:
> * **High Waste:** On average, it takes **33.33 sampling attempts** to generate a single valid incorrect response.
> * **Failure Cases:** For **16%** of problems, Step-DPO fails to find *any* error even after 100 attempts.
>
> > **Implication:** The vast majority (>95%) of tokens generated by Step-DPO are discarded "failed attempts," leading to massive computational waste.
>
> **Crucially, "hard-to-sample" does not mean "unimportant."** In real-world deployments at scale (e.g., widely-used services like ChatGPT serving billions of requests monthly), even a seemingly negligible error rate (e.g., 1%) translates to **tens of millions of failures** in absolute terms. Sampling-based methods effectively "give up" on these rare failure modes because they are statistically invisible during standard training. KEEP can significantly **alleviate the long tail distribution problem** of error types based on sampling methods through active error exploration.
>
> ## **(b.3) Theoretical & Empirical Cost Comparison (10x Reduction)**
> Appendix B derives that theoretically, the baseline consumes at least **2.25x** more tokens than KEEP (Eq. 16). In practice, due to the long-tail distribution of sampling difficulty, the gap is much wider.
>
> We provide a detailed breakdown of the token costs based on the complete statistics reported in the following table.
>
> **Table. The results of computational costs based on the number of input and output tokens, assuming that one QA pair needs to generate one preference pair data, the following is the average results for one QA pair.**
> | Method | Process | Model | Token Costs |
> | :--- | :--- | :--- | :--- |
> | **Step-DPO** | Sampling of Incorrect Answers | target LLM | 4,329.08 |
> | | Error Step Localization | GPT-4o | 566.94 |
> | | Sampling of Correct Answers | target LLM | 3,876.72 |
> | | **Total** | |  **8,772.74** |
> | **KEEP** | Key Step Identification | target LLM | 284.07 |
> | | Proactive Error Exploration | GPT-4o |  157.85 |
> | | Speculative Filtering | target LLM | 467.88 |
> | | **Total** | | **909.80** |
>
> **Observations:**
> 1.  **Massive Sampling Overhead:** As shown in the *Description* column, Step-DPO requires inputting the question "multiple times" to sample incorrect/correct responses, leading to an enormous consumption (~8,200+ tokens) purely on repeated sampling.
> 2.  **10x Reduction:** KEEP reduces the total token consumption from **~8,772** to **~910**, representing a **~9.6x reduction**.
> 3.  **GPT-4o Efficiency:** Even regarding the usage of the external teacher model (GPT-4o), KEEP's targeted *Proactive Error Exploration* (157.85 tokens) is significantly more efficient than Step-DPO's *Error Step Localization* (566.94 tokens).
>
> **Conclusion:** KEEP is not an additional burden; it is a computational optimization that replaces "brute-force sampling" with efficient "targeted editing."
>
>
> ---

---

> ### Author Response · Authors · 2025-11-24
> **Response to Reviewer 6Sce (part 3)**
>
> # **(c) Is KEEP more effective than instance-level sampling?**
>
> **Response: KEEP Consistently Outperforms Sampling Baselines with Better Pareto Efficiency**
>
> Finally, we address the effectiveness concern. We compared KEEP against standard sampling-based methods (Step-DPO and SCDPO) across 6 benchmarks.
>
> ## **(c.1) Performance Comparison: Superior Accuracy Across Models**
>
> As shown in the following table, KEEP consistently outperforms baselines on both in-domain and out-of-domain datasets.
>
> ### **Table. Accuracy comparison on Qwen2-7B and Llama-3.1-8B**
>
> | Backbone | Method | GSM8K | MATH | AQuA | SVAMP | AIME24 | Odyssey |
> | :--- | :--- | :---: | :---: | :---: | :---: | :---: | :---: |
> | **Qwen2-7B** | Step-DPO | 88.5 | 55.8 | 63.0 | 88.7 | 2/30 | 34.0 |
> | | SCDPO | 86.8 | 51.6 | 70.5 | 89.6 | 1/30 | 18.9 |
> | | **KEEP** | **89.1** | **59.9** | **72.8** | **91.9** | **4/30** | **44.4** |
> | **Llama-3.1-8B**| SCDPO | 84.4 | 48.8 | 63.2 | 85.1 | 2/30 | 48.5 |
> | | **KEEP** | **88.3** | **52.0** | **65.4** | **88.2** | **4/30** | **54.3** |
>
>
> **Observation:**
> * **Broad Improvements:** KEEP achieves the highest accuracy across all evaluated datasets on both backbones. Notably, on **Odyssey-MATH**, KEEP surpasses SCDPO by over **25%**.
> * **Consistent Gains on Llama:** On Llama-3.1-8B, KEEP outperforms SCDPO by **+3.2%** on MATH and **+3.9%** on GSM8K, proving its effectiveness is not limited to the Qwen family.
> * **Pareto Efficiency:** KEEP delivers **higher performance** while consuming **~10x fewer tokens**, establishing a superior Pareto frontier compared to instance-level sampling.
>
>
> ## **(c.2) Error Quality: Avoiding "Mode Collapse"**
>
> Why is KEEP more effective? Appendix G (Figures 12-14) analyzes the error distribution:
> * **Baselines (Mode Collapse):** Sampling-based methods tend to over-generate simple errors (e.g., calculation slips), leading to a "long-tail" distribution where complex logical errors are rare.
> * **KEEP (Diverse & Structured):** By actively exploring errors on key steps, KEEP produces a balanced mix of errors, including condition misinterpretations and reasoning jumps. These "harder" negatives provide richer gradients for the preference model.
>
> ---
>
> # **Summary of Response to Weakness 1**
>
> To wrap up our response regarding the Draft Model, Cost, and Effectiveness:
>
> 1.  **OOD Risk (Mitigated):** We do not blindly accept Draft Model outputs. By applying **Speculative Filtering**, we remove ~60-90% of high-OOD-risk samples. The remaining errors are "potential negatives" compatible with the policy model.
> 2.  **Computational Cost (Reduced):** Contrary to intuition, using a Draft Model is *cheaper* than the baseline because it eliminates the massive waste of invalid sampling. Theoretical and empirical evidence confirms a **~10x reduction** in target-model tokens.
> 3.  **Effectiveness (Verified):** KEEP consistently outperforms sampling baselines (e.g., +25% on Odyssey vs SCDPO) by generating more diverse, pedagogically valuable errors.
>
> We hope this comprehensive analysis alleviates the reviewer's concerns. We are happy to provide further clarifications if needed.
>
> ---

---

> ### Author Response · Authors · 2025-11-24
> **Response to Reviewer 6Sce (part 4)**
>
> # **Response to Weakness 2**
> # **Novelty of Key-step Identification**
>
> We thank the reviewer for this thoughtful comment. We agree that Perplexity (PPL) and Entropy are standard metrics. However, our contribution is not utilizing them in isolation, but identifying that **their linear fusion serves as a computationally efficient proxy for "Expected Information Gain,"** which is otherwise prohibitively expensive to compute.
>
> We address the novelty concern from three dimensions: **Theoretical Grounding**, **Complementary Mechanisms**, and **Systemic Efficiency**.
>
> ## **2.1 Theoretical Novelty: A Tractable Proxy for Information Gain**
>
> In **Appendix D**, we provide a theoretical derivation proving that our scoring rule is not an arbitrary heuristic, but a grounded approximation of the **Expected Information Gain (IG)** regarding the final answer's correctness.
>
> * **Theoretical Link (Theorem D.1 & D.2):**
>     * **Theorem D.1** proves that steps with high PPL (low probability) offer the mathematically largest potential for log-likelihood improvement if corrected.
>     * **Theorem D.2** proves that steps with high Conditional Entropy serve as an upper bound for the Mutual Information between the step and the final outcome.
> * **The "Proxy" Insight:** calculating the true Information Gain requires expensive Monte-Carlo rollouts (thousands of times slower). Our fusion score, $\alpha \cdot \text{norm}(\text{PPL}) + \beta \cdot \text{norm}(\text{Entropy})$, mathematically acts as a **low-cost variational proxy** for this value.
> * **Empirical Verification (Figure 4):** As shown in **Figure 4** (Appendix D), we conducted Monte-Carlo simulations to measure the *true* empirical Information Gain ($EI_{MC}$). The results show a statistically significant correlation (**Spearman’s $\rho=0.488$, $p < 10^{-22}$**) between our lightweight proxy and the expensive ground truth.
>
> **Novelty Claim:** The innovation is establishing that **simple linear fusion** effectively approximates **complex causal importance**, allowing us to identify critical steps without expensive rollouts.
>
> ## **2.2 Methodological Novelty: Complementary Capture of "Hardness" and "Uncertainty"**
>
> In **Appendix E**, we analyze why neither metric suffices alone. They capture fundamentally different types of reasoning failures, and using only one leads to suboptimal selection (as verified in our ablation study).
>
>
> * **PPL (Inherent Complexity):** Reflects the model's unfamiliarity with the specific step content, such as calculation bottlenecks. As shown in **Figures 6, 7 and 8** (Appendix E), high-PPL steps often concentrate on complex operational steps.
> * **Entropy (Contextual Uncertainty):** Reflects the informational value of a step within the reasoning path. As shown in **Figures 6, 7 and 8** (Appendix E), high-entropy steps often capture critical logical transitions or initial problem-solving approaches.
>
>
> **Evidence:** We conducted an ablation study to isolate the contribution of the fusion strategy.
>
> **Table. Ablation of Key-step Identification Strategies (Accuracy %)**
>
> |Backbone|Key-step Method|GSM8K|MATH|Avg|
> |:---|:---|:---:|:---:|:---:|
> |**Qwen2-7B**|PPL Only|88.1|57.8|73.0|
> ||Entropy Only|87.9|57.5|72.7|
> ||**KEEP (Fusion)**|**89.1**|**59.9**|**74.5**|
> |**Llama-3.1-8B**|PPL Only|85.8|48.0|66.9|
> ||Entropy Only|85.5|47.5|66.5|
> ||**KEEP (Fusion)**|**88.3**|**52.0**|**70.2**|
>
>
> **Conclusion:**
> The fusion strategy consistently yields substantial gains over single metrics.
> * On **Qwen2-7B**, the fusion improves over single metrics by **+2.1%** on MATH.
> * On **Llama-3.1-8B**, the gap is even more pronounced, with fusion outperforming single metrics by **+4.0%** on MATH.
>
> This confirms that identifying a "Key Step" requires simultaneously detecting *where the model is confused* (Entropy) and *where the model is weak* (PPL). Neither signal alone is sufficient to capture the full spectrum of reasoning errors, especially for weaker base models like Llama-3.1-8B where error patterns are more diverse.
>
> ## **2.3 Systemic Novelty: The "Gatekeeper" for Efficiency**
>
> Finally, the novelty of this module must be viewed within the full **KEEP pipeline**. It is not just a scoring rule; it is the **structural gatekeeper** that enables the efficiency claims discussed in Weakness 1.
>
> * **Search Space Reduction:** By filtering out non-critical steps (bottom 50%), this module reduces the search space for Error Exploration by half.
> * **Resource Allocation:** It directs the "Draft Model" budget only to steps where errors are most likely to impact the final answer, rather than wasting resources on trivial steps.
>
> **Summary:**
> While PPL and Entropy are standard tools, KEEP’s contribution is **deriving their fused utility from information theory** (Appendix D) and deploying them as a **cost-effective selector** to enable scalable preference optimization. This transforms them from simple "metrics" into a core "active learning" component.
>
> ---

---

> ### Author Response · Authors · 2025-11-24
> **Response to Reviewer 6Sce (part 5)**
>
> # **Response to Weakness 3**
> # **Why DPO is a Principled Choice and KEEP's Compatibility with RL**
>
> We sincerely thank the reviewer for this insightful suggestion. We fully agree that step-wise reasoning signals are naturally aligned with process-supervision RL methods (like PPO and GRPO). We clarify our design choice from two perspectives: **(1) Structural Alignment & Efficiency** and **(2) Generality for RL Pipelines**.
>
> ## **3.1 Why KEEP (Step-wise) + DPO is a principled and effective design choice**
>
> ### **(1) 1-to-1 Structural Match**
> KEEP’s data construction pipeline explicitly outputs **fine-grained preference pairs**:
> * A **chosen** step (correct local reasoning)
> * A **rejected** step (carefully constructed plausible error)
>
> This structure naturally aligns with the DPO objective, which optimizes the policy directly from preference pairs. Crucially, by leveraging such high-quality granular data, **KEEP effectively elevates the DPO paradigm—traditionally limited to coarse response-level feedback—to a precise step-wise optimization**, ensuring that the model learns to discriminate reasoning logic at critical turning points on multi-step reasoning tasks such as mathematics.
>
> ### **(2) Addressing the "Inefficiency" Concern**
> The reviewer mentioned that DPO might be "inefficient." We respectfully argue that for *preference learning*, DPO is often **more efficient** than RL in terms of computational stability:
> * **No Rollouts:** DPO training does not require online generation (rollouts) during training, reducing computational cost significantly compared to PPO/GRPO.
> * **Local Correction:** Math errors are often local (e.g., a single sign flip). Step-wise DPO  allows the model to focus gradients precisely on the erroneous token sequence, offering a direct signal that response-level DPO often struggles to isolate.
>
> ### **(3) Empirical Competitiveness**
> In our main paper (Table 5), we compared KEEP (Step-wise DPO) directly against PPO and GRPO baselines.
> * **Result:** KEEP achieves **89.1%** on GSM8K and **59.9%** on MATH, matching or outperforming standard RLVR methods.
> * **Conclusion:** This empirically proves that the Step-wise DPO formulation is highly effective and not inherently "inefficient" for this task.

---

> ### Author Response · Authors · 2025-11-24
> **Response to Reviewer 6Sce (part 6)**
>
> ## **3.2 KEEP Enhances the RL Ecosystem (Validation of Reviewer's Insight)**
>
> We deeply appreciate the reviewer's insightful suggestion regarding the integration of KEEP with RL frameworks (like PPO or GRPO).
>
> **Why PRM?** We explicitly recognize that the most principled and effective way to integrate KEEP with RL is by enhancing the **Reward Model (RM)**. Since the Reward Model serves as the **core component** guiding RL algorithms, improving the RM directly translates to better PPO/GRPO performance. KEEP, capable of generating high-quality, fine-grained step-level preference pairs, is naturally poised to optimize advanced **Process Reward Models (PRMs)**.
>
> To validate this, we utilized KEEP data to train PRMs and evaluated them on **ProcessBench**, a widely used benchmark for assessing process supervision capabilities. The experiments covered four diverse datasets, and the results confirm that KEEP-generated data significantly improves the model's ability to distinguish correct from incorrect steps.
>
> ### **Table. Performance of PRMs trained on KEEP data vs. Baselines**
>
> | Model |  | GSM8K | |  | MATH | |  | Olympiad | |  | Omni-MATH | |
> | :--- | :---: | :---: | :---: | :---: | :---: | :---: | :---: | :---: | :---: | :---: | :---: | :---: |
> |  | **Error** | **Correct** |**F1** | **Error** | **Correct** |**F1** |**Error** | **Correct** |**F1** |**Error** | **Correct** |**F1** |
> | PRM | 33.8 | **99.0** | 50.4 | 21.7 | 72.2 | 33.4 | 8.2 | 43.1 | 13.8 | 9.6 | 45.2 | 15.8 |
> | PRM+SPO | 38.6 | 98.4 | 55.5 | 20.4 | 74.3 | 31.8 | 8.3 | 57.5 | 14.5 | 9.2 | 41.0 | 15.1 |
> | PRM+SCDPO | 33.8 | 98.4 | 50.3 | 22.4 | 70.7 | 34.0 | 8.6 | 40.1 | 14.2 | 10.3 | 42.3 | 16.5 |
> | **PRM+KEEP** | **43.5** | 96.4 | **59.9** | **22.4** | **79.8** | **35.0** | **8.8** | **63.1** | **15.4** | **10.4** | **54.4** | **17.5** |
>
> *(Note: "Error" denotes accuracy on incorrect steps, "Corr" on correct steps. KEEP consistently improves identification of errors, which is the hardest part of process supervision.)*
>
> **Observation:**
>
>   * **Comprehensive Improvement:** The PRM trained on KEEP data achieves the **highest F1 scores** across all four benchmarks in ProcessBench.
>   * **Superior Error Detection:** Crucially, KEEP significantly boosts the **Error Accuracy** (e.g., **43.5 vs 33.8** on GSM8K), indicating it provides much richer signals for "potential negatives" compared to baselines.
>   * **Robustness:** Even on out-of-domain datasets like OlympiadBench and Omni-MATH, KEEP maintains its lead.
>
> **Conclusion:**
> This validates the reviewer's intuition: KEEP's high-quality, fine-grained data is a versatile asset. While we chose DPO for its simplicity and efficiency in this work, **KEEP can serve as a powerful "Data Engine" to power mainstream RL/Process Supervision pipelines.**
>
> ---
>
> ### **Summary for Weakness 3**
>
> 1.  **Choice Justification:** We chose step-wise DPO because it structurally matches our data format (preference pairs) and offers a stable, rollout-free training efficiency.
> 2.  **RL Compatibility:** We empirically demonstrate that KEEP data trains superior Process Reward Models, confirming that our method contributes to the process supervision ecosystem , as suggested by the reviewer.
> 3. **Future Potential:** We agree that deeper integration with RL is a promising direction; for instance, leveraging KEEP's PPL and Entropy metrics to guide exploration intensity on critical steps during PPO/GRPO training represents an exciting avenue for future work.
>
>
> ---
>
>
> **We humbly hope our response has addressed your concerns. If you have any additional concerns or comments that we may have missed in our responses, we would be most grateful for any further feedback from you to help us further enhance our work.**

---

### Note · Program_Chairs · 2026-01-17
**Submission Desk Rejected by Program Chairs**

The following references in this submission do not refer to real documents and/or have major errors in bibliographic information:

 Alex Mallen, Alexandru Tifrea, and Chitta Baral. "The zebra puzzle: A benchmark for structured logical reasoning." arXiv preprint arXiv:2305.17631, 2023.